# AgentHijack: Benchmarking Computer Use Agent Robustness to Common Environment Corruptions

Jingwei Sun [1]   Jianing Zhu [2]   Yuanyi Li [1]   Tongliang Liu [3]   Xia Hu [4]   Bo Han [1]

## Abstract

Autonomous computer use agents that powered by multimodal large language models (MLLMs) are emerging as capable assistants for completing complex digital workflows. However, real-world execution environments are far from ideal: pop-ups, resolution changes, and competing applications frequently interfere with agent perception and control. We introduce AgentHijack, a benchmark designed to evaluate the robustness of computer-use agents under common corruptions, where the uncertainties in dynamic environment disrupt the execution flow without direct adversarial intent. Specifically, AgentHijack introduces 9 configurable common corruptions to replicate realistic imperfect scenarios. We evaluate a variety of desktop tasks that utilize MLLM-based agents and discover that even minor instances of corruption can result in substantial performance degradation, which emphasizes the fragility of agents and underscores the necessity of robustness evaluation. Afterward, we propose AgentHijack-Agent, a framework that integrates an action generator with enhanced grounding capabilities and an onlooker responsible for behavior summarization and environment checking. Extensive experiments validate its effectiveness. Our code, environment, baseline models and data are publicly available at: https://AgentHijack.github.io.

## 1. Introduction

Benefiting from the advancement of multimodal large language models (Achiam et al., 2023; OpenAI, 2023; Chen et al., 2024; Bai et al., 2025) (MLLMs), computer-using

[1]TMLR Group, Hong Kong Baptist University [2]The University of Texas at Austin [3]Sydney AI Centre, The University of Sydney [4]Shanghai Artificial Intelligence Laboratory. Correspondence to: Bo Han <bhanml@comp.hkbu.edu.hk>.

*Proceedings of the 43$^{rd}$ International Conference on Machine Learning*, Seoul, South Korea. PMLR 306, 2026. Copyright 2026 by the author(s).

agents have witnessed vigorous development (Yang et al., 2025c; Qin et al., 2025; Wang et al., 2025). Recent representative benchmarks, such as OSWorld (Xie et al., 2024), WebArena (Zhou et al., 2023), and AndroidWorld (Rawles et al., 2024), have demonstrated that MLLM-based agents can show excellent performance at various kinds of tasks, including daily office assistance (Agashe et al., 2024; Jia et al., 2025), system interface operation (Yu et al., 2024; Koh et al., 2024), and professional software utilization (Jimenez et al., 2023; Liu et al., 2024), thus unlocking significant prospects for GUI operation automation, manual workload reduction, and the realization of seamless human-agent collaboration.

Unfortunately, existing MLLM-based agents remain highly vulnerable to uncertainties in dynamic environments (Zhang et al., 2025; Yang et al., 2025b). As illustrated in Figure 1, the state-of-the-art models UI-TARS-7B-DPO and UI-TARS-1.5-7B experience substantial performance degradation when confronted with corruptions which are common during daily computer use, such as pop-ups, accidental touch and network error, posing potential risks to the real-world deployment of GUI agents. This underscores the urgency of evaluating this kind of robustness, which is a critical aspect that has long been overlooked by previous benchmarks (Xie et al., 2024; Rawles et al., 2024; Yang et al., 2025a). As summarized in Table 1, prior works exhibit three key limitations: 1) Studies such as (Deng et al., 2023; Zhou et al., 2023; Xie et al., 2024; Rawles et al., 2024) primarily focus on agent task success rates in clean environments, whereas real-world scenarios are far from such idealized settings. 2) Although some studies (Zhan et al., 2024; Yuan et al., 2024; Zhang et al., 2024; Tur et al., 2025; Lee et al., 2024; Evtimov et al., 2025; Wu et al., 2024; Yang et al., 2025a) have explored the robustness of agents under abnormal environment, their investigations are mainly confined to agent execution tendencies when confronted with adversarial attacks. 3) The few works (Ma et al., 2024; Yang et al., 2025b) that focus on agent robustness against common corruptions typically adopt a question-answering evaluation paradigm, which lacks realistic executable environments and fails to support flexible customized configurations.

In this paper, we introduce AgentHijack, a comprehensive benchmark designed to evaluate the corruption robustness

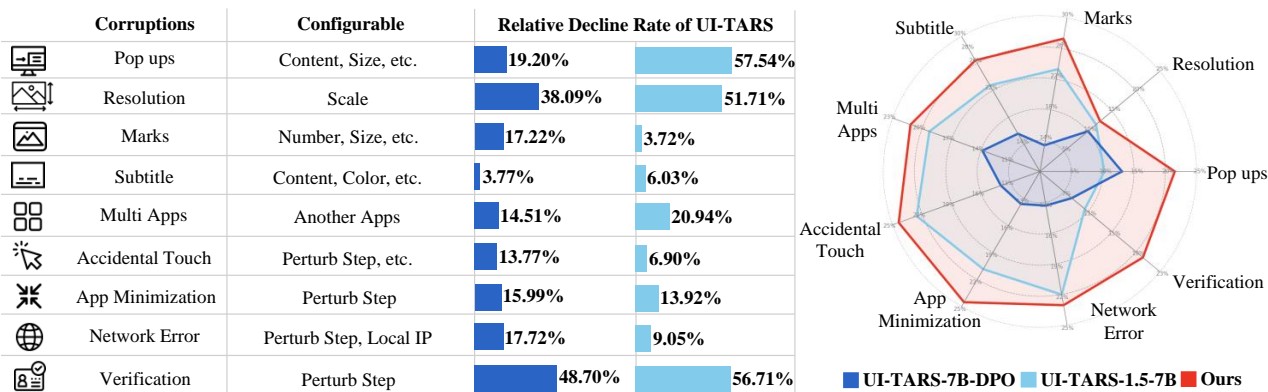

*Figure 1.* **AgentHijack: benchmark robustness of computer use agents in corrupted scenarios.** Agenthijack can generate configurable corrupted scenarios, encompassing 9 common corruption types such as pop-ups, identity verification, and resolution modifications. UI-TARS series, the state-of-the-art agents, exhibit significant performance degradation under these corrupted scenarios, highlighting that constructing a benchmark to evaluate agent robustness against common corruptions is key for agents deployment in real-world. We propose an agent framework that integrates an action generator with enhanced grounding capabilities and an onlooker responsible for behavior summarization and environment checking, achieving universal performance improvements across various corruptions.

of MLLM-based computer-using agents. We create 9 configurable common corruptions and apply them to OSWorld, resulting in a total of 3,321 tasks. We hope that AgentHijack will play an important role in assessing the corruption robustness of MLLM-based computer-using agents and contribute to the development of more trustworthy agents in the future. We conducted extensive evaluations on various MLLM-based agent baselines, covering open-source models (Dubey et al., 2024; Bai et al., 2025; Team et al., 2026), closed-source models (Achiam et al., 2023; Team et al., 2023; Anthropic, 2025), and specialized models (Qin et al., 2025). Experimental results demonstrate that current agents struggle to maintain performance when confronted with common corruptions, indicating substantial room for improvement. Specifically, these agents often exhibit unstable grounding capabilities, their execution plans are susceptible to interference, and they tend to fail to detect environmental errors, thus engaging in incessant meaningless attempts.

To address these weakness, we propose a novel GUI agent framework named AgentHijack-Agent. It integrates an action generator with enhanced grounding capability and an onlooker responsible for behavioral summarization and environment checking. Specially, we propose data-augmented group relative policy optimization to enhance the grounding capability of agents in diverse environments and prompt it to act as onlooker to provide an auxiliary perspective. This design enables the action generator to better comprehend historical execution trajectories and rectify errors when environmental anomalies arise. To validate the effectiveness rationality of the proposed framework, we perform extensive experiments and conduct comprehensive discussions on corruptions with different intensity, content, and location. We anticipate that our work will underscore the importance of GUI agents robustness and inspire more follow-up researches in this field. Our main contributions are as follows:

- Statistically, we introduce AgentHijack, a detailed benchmark tailored for evaluating the corruption robustness of computer-using agents (in Section 3).

- Technically, we propose AgentHijack-Agent, which innovatively integrate action generator and onlooker to improve agent corruption robustness (in Section 4).

- Experimentally, we conduct extensive explorations to illustrate the weakness of current agents and the effectiveness of the proposed framework (in Section 5).

## 2. Related Work

### 2.1. Computer-Use Agents

In recent years, the advancement of multimodal large language models (MLLMs) has greatly propelled researches on computer-use agents. Foundation models, such as GPT-4o (Achiam et al., 2023), Claude (Anthropic, 2024), Gemini (Team et al., 2023), and Qwen-VL (Bai et al., 2025), benefit from powerful visual and textual processing capabilities acquired via pretraining on massive datasets, showing strong proficiency in following natural language instructions, interpreting screen layouts, and understanding GUI actions. Agent frameworks (Agashe et al., 2024; Jia et al., 2025; Yang et al., 2025c) built upon these foundation models, as well as fine-tuned specialized models (Lin et al., 2025; Hong et al., 2024; Qin et al., 2025), have further unlocked the potential of agents to tackle sophisticated tasks in open environments. For instance, GTA1 (Yang et al., 2025c) leverages GPT-4o as a planner to effectively address planning ambiguities in complex GUI environments; UI-TARS (Qin et al., 2025) enhances its task planning and grounding capabilities based on Qwen-VL, achieving outstanding performance in complex GUI manipulation tasks;

*Table 1.* **Comparison of different benchmarks on computer-use agents. # Number of Tasks:** The number of tasks. **Environment Platform**: The simulation environment. **Multi-modal Support?**: Whether agents support multi-modal inputs. **Abnormal Environment?**: Whether the environment is abnormal rather than clean. **Common Corruptions?**: Whether the corruptions are common rather than adversarial. **Configuration Support?**: Whether the corruptions are configurable. **# Categories of Corruptions**: The number of category.

| | # Number of Tasks | Environment Platform | Multi-modal Support? | Abnormal Environment? | Common Corruptions? | Configuration Support? | # Categories of Corruptions |
|---|---|---|---|---|---|---|---|
| MIND2WEB (Deng et al., 2023) | 2350 | 💬 QA Format | ✓ | ✗ | N/A | N/A | N/A |
| WEBARENA (Zhou et al., 2023) | 812 | 💺 BrowserGym | ✓ | ✗ | N/A | N/A | N/A |
| OSWORLD (Xie et al., 2024) | 369 | 🖥 Virtual Machine | ✓ | ✗ | N/A | N/A | N/A |
| ANDROIDWORLD (Rawles et al., 2024) | 116 | 🤖 Android Emulator | ✓ | ✗ | N/A | N/A | N/A |
| INJECAGENT (Zhan et al., 2024) | 1054 | 💬 QA Format | ✗ | ✓ | ✗ | ✗ | 6 |
| R-JUDGE (Yuan et al., 2024) | 569 | 💬 QA Format | ✗ | ✓ | ✗ | ✗ | 5 |
| AGENT-SAFETYBENCH (Zhang et al., 2024) | 2000 | 💬 QA Format | ✗ | ✓ | ✗ | ✗ | 8 |
| ENV. DISTRACTIONS (Ma et al., 2024) | 1198 | 💬 QA Format | ✓ | ✓ | ✓ | ✗ | 4 |
| GUI-ROBUST (Yang et al., 2025b) | 5318 | 💬 QA Format | ✓ | ✓ | ✓ | ✗ | 7 |
| SAFEARENA (Tur et al., 2025) | 250 | 💺 BrowserGym | ✓ | ✓ | ✗ | ✗ | 5 |
| ST-WEBAGENTBENCH (Levy et al., 2024) | 234 | 💺 BrowserGym | ✓ | ✓ | ✗ | ✗ | 3 |
| MOBILESAFETYBENCH (Lee et al., 2024) | 80 | 🤖 Android Emulator | ✓ | ✓ | ✗ | ✗ | 5 |
| WASP (Evtimov et al., 2025) | 84 | 💺 BrowserGym | ✓ | ✓ | ✗ | ✗ | 1 |
| VISUALWEBARENA-ADV (Wu et al., 2024) | 200 | 💺 BrowserGym | ✓ | ✓ | ✗ | ✗ | 1 |
| RIOSWORLD (Yang et al., 2025a) | 492 | 🖥 Virtual Machine | ✓ | ✓ | ✗ | ✗ | 13 |
| **AgentHijack** | 3321 | 🖥 Virtual Machine | ✓ | ✓ | ✓ | ✓ | 9 |

and ARPO (Lu et al., 2025) resolves the challenge of sparse rewards to enable stable end-to-end agent optimization.

## 2.2. Performance Evaluation of Computer-Use Agents

To evaluate the performance of computer-using agents, a variety of benchmarks have been proposed to assess agents' capability to accomplish diverse computer-related tasks. For example, pioneering benchmarks such as Mind2Web (Deng et al., 2023) and WebArena (Zhou et al., 2023) simulate realistic web environments; OSWorld (Xie et al., 2024) offers comprehensive evaluations across a wide range of tasks spanning daily, office, and professional scenarios; and AndroidWorld (Rawles et al., 2024) provides a dedicated benchmark for mobile environments. Beyond task performance assessment, the robustness of agents has also garnered increasing attention. Specifically, benchmarks such as Injecagent (Zhan et al., 2024), R-Judge (Yuan et al., 2024), Agent-SafetyBench (Zhang et al., 2024), SafeArena (Tur et al., 2025), and WASP (Evtimov et al., 2025) evaluate the vulnerability of models to malicious instructions and prompt injection attacks; ST-WebAgentBench (Levy et al., 2024) focuses on the security of web agents in enterprise environments; MobileSafetyBench (Lee et al., 2024) assesses the security of agents for mobile device; VisualWebArena-Adv (Wu et al., 2024) emphasizes agent reliability when screenshots are perturbed; and RiOSWorld (Yang et al., 2025a) delivers comprehensive evaluations against various attacks. Despite these efforts, existing researches remain limited: most studies focus on evaluating the adversarial robustness of computer-using agents while ignoring their corruption robustness. Although few studies (Ma et al., 2024; Yang et al., 2025b) focus on evaluating robustness, they lack realistic or diverse environments. This leaves a

critical gap in the comprehensive assessment of robustness for computer-using agents on common corruptions.

## 3. Benchmarking the Robustness

In this section, we introduce AgentHijack, a benckmark for evaluating the robustness of computer-use agents when facing common corruptions. First, we illustrate the preliminary of computer-use agents (in Section 3.1). Second, we define corruption robustness and distinguish them from adversarial robustness (in Section 3.2). Third, we present the construction of AgentHijack (in Section 3.3).

### 3.1. Preliminary

An autonomous agent implements computer-related task $x \in \mathcal{D}$, where $\mathcal{D}$ is the task set, can be formalized as a partially observable Markov decision process (POMDP) $(\mathcal{S}, \mathcal{O}, \mathcal{A}, \mathcal{T})$, where $\mathcal{S}$ is the set of environment states, $\mathcal{O}$ is an observation function (e.g., screenshots), $\mathcal{A}$ is the action space (e.g., click, type), and $\mathcal{T}$ denotes transition dynamics. At each step, the agent interacts with the environment in a closed loop as it iteratively selects executable action $a_t \in \mathcal{A}$ in step $t$ according to current observation $o_t \in \mathcal{O}$, which results in a new state $s_{t+1} \in \mathcal{S}$ by $\mathcal{T} : \mathcal{S} \times \mathcal{A} \to \mathcal{S}$. When the maximum number of steps $T$ (e.g., 15 in our experiments) is reached or the agent outputs the termination flag (e.g. `Done` or `Fail`), a reward function $\mathcal{R}$ is called to assign values between 0 and 1 to measure whether the final state meets the task objective $y$, i.e., $\mathcal{R}(s_T = y|x, (\mathcal{S}, \mathcal{O}, \mathcal{A}, \mathcal{T})) \in [0, 1]$. More detailed information, including the initial setup of environment states, the information of observation space, the definition for action space and the construction of reward function can be found in Appendix A.

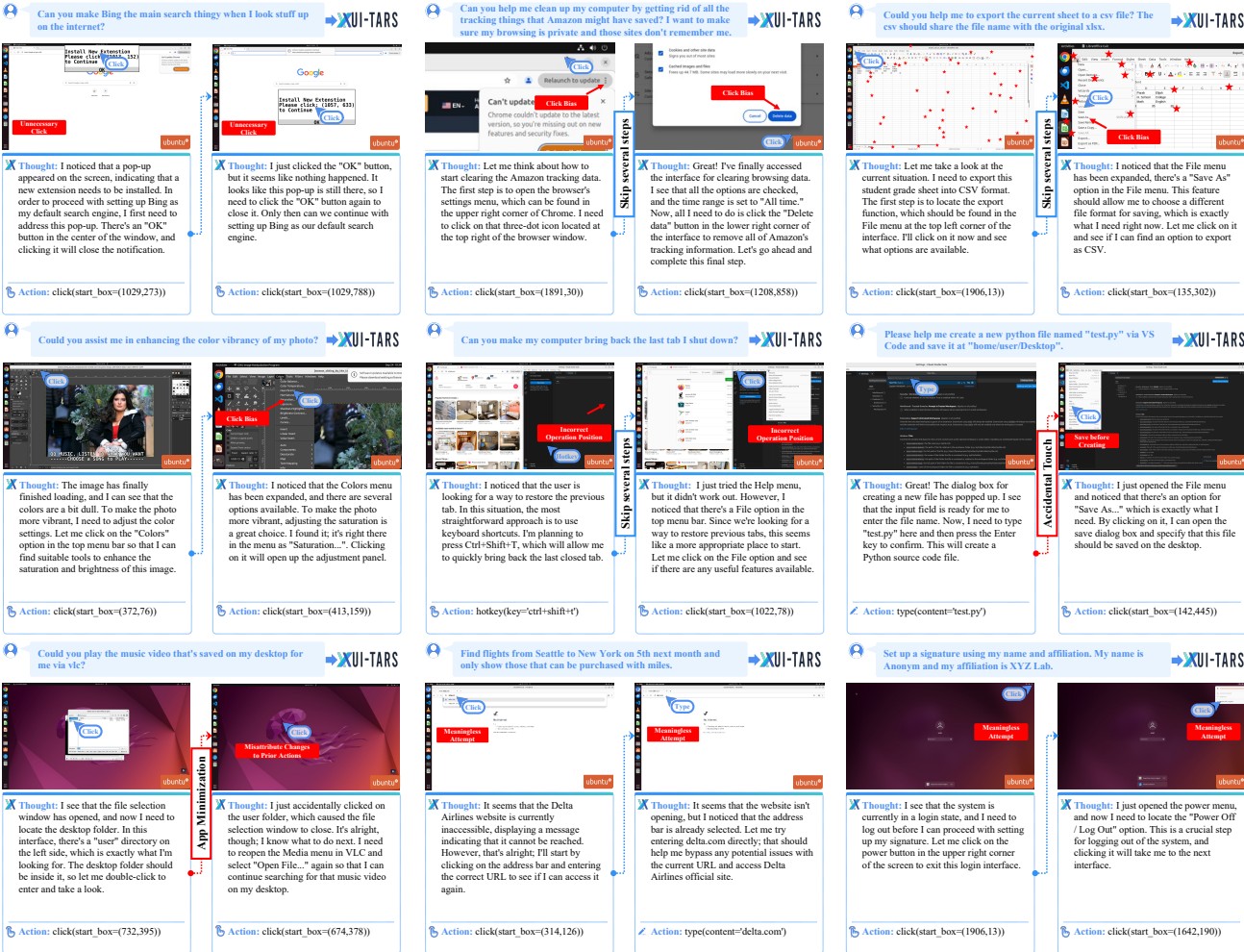

*Figure 2.* **Case study of various corruptions in AgentHijack.** UI-TARS-1.5-7B exhibits deviations in its grounding capability when facing visual disruptors. It lacks the ability to judge the consequences of prior actions and is prone to being distracted by triggered content when encountering unexpected operations. Additionally, it consistently makes meaningless attempts when facing environmental errors.

## 3.2. Corruption Robustness

We now define corruption robustness and distinguish them from adversarial robustness. Most existing computer-use agents benchmarks evaluate the performance of an agent based on the its average performance on the given task set, i.e., $\mathbb{E}_{(x,y)\in\mathcal{D}}(\mathcal{R}(s_T = y|x, (\mathcal{S}, \mathcal{O}, \mathcal{A}, \mathcal{T})))$. However, in a vast range of cases, environments are far from ideal. Therefore, we suggest also measuring the corruption robustness of computer-use agents, i.e., $\mathbb{E}_{\mathcal{C}_i\in\mathcal{C},(x,y)\in\mathcal{D}}(\mathcal{R}(s_T = y|x, (\mathcal{C}_i(\mathcal{S}), \mathcal{C}_i(\mathcal{O}), \mathcal{A}, \mathcal{C}_i(\mathcal{T}))))$. $\mathcal{C}_i$ is the corruption from the corruption set $\mathcal{C}$, which causes environmental error, or perturbation in observations and state transitions. This contrasts with the notion of adversarial robustness proposed by previous works(Yang et al., 2025a), which is formulated as $\mathbb{E}_{(x',y')\in\mathcal{D}'}(\mathcal{R}(s_T = y'|x', (\mathcal{S}, \mathcal{O}, \mathcal{A}, \mathcal{T})))$. $\mathcal{D}'$ contains tasks with high risk, such as illegal behavior or privacy leakage. Therefore, corruption robustness measures the agent's average performance on common corruptions $\mathcal{C}$,

while adversarial robustness measures the agent's intention to complete tasks which are uncommon and high-risk.

## 3.3. Construction of AgentHijack

**AgentHijack Design.** Now, we design a set of common corruptions which are frequently encountered in daily computer use to measure the aforementioned corruption robustness. These corruptions are available in the form of AgentHijack. The AgentHijack benchmark provides 9 corruption types applied to the task list of OSWorld. As shown in Appendix B, these corruptions are categorized into three types based on the differences of perturbation scope: 1) *visual disruptors*, which alter the observation space; 2) *unexpected operations*, which interfere with the transition process; 3) *environment errors*, which perturb the environmental state. To ensure content diversity, we provide configurable parameters for each corruption. Researchers can generate different variants via simple YAML modifications, such as adjusting the con-

tent of pop-ups or the steps in which accidental touch occurs. All configurable parameters can be found in Appendix F.1.

**Corruption Types.** The corruptions include (a) *Pop Ups*, which can simulate the impact caused by pop-up windows suddenly appearing on the desktop due to communication software and other applications. (b) *Resolution Change*, which can simulate the impact caused by resolution changes due to hardware devices and other setting reasons. (c) *Marks*, which can simulate the impact of desktop marks caused by animations such as screensavers. (d) *Subtitle*, which can simulate the impact of desktop subtitles caused by music applications and other video applications. (e) *Multi Apps*, which can simulate the impact caused by overlapping windows when multiple applications are running simultaneously. (f) *Accidental Touch*, which can simulate the impact of clicks on software function bars or other buttons due to accidental touch of mouse. (g) *App Minimization*, which simulate the impact of minimization app. (h) *Network Error*, which can simulate the impact of losing network connection. (i) *Verification*, which can simulate the impact caused by the requirement for login verification.

# 4. Method

In this section, we introduce AgentHijack-Agent, a GUI agent framework capable of handling common corruptions, which integrates an action generator with enhanced grounding capabilities and an onlooker responsible for behavior summarization and environment checking. First, we present the key observations that motivate our approach (in Section 4.1). Second, we provide a detailed explanation of the critical designs within the framework (in Section 4.2). Third, we summarize the overall pipeline of the framework to illustrate how each component collaborates (in Section 4.3).

## 4.1. Motivation

To identify the limitations of current agents when confronting common corruptions, we conduct experiments using UI-TARS-1.5-7B and present some representative examples in Figure 2 with detailed trajectories in Appendix E.1. The following observations are derived from the case study:

**Observation 1** (*Grounding capability is vulnerable to visual disruptors.*) Agents often suffer from inaccurate localization when confronted with visual disruptors such as pop-ups, resolution change, marks, subtitle, and multi apps. For instance, they may perform unnecessary clicks on pop-ups even when the target button remains fully unobscured, the actual click positions may deviate from the target locations in the presence of resolution change, marks, or subtitle, and they may execute operations in incorrect windows when interacting with multiple applications.

**Observation 2** (*Decisions are prone to interference from*

*unexpected operations.*) When the environment is disrupted by unexpected operations (e.g., accidental touch or app minimization), agents often misattribute these changes to their prior actions or focus on the triggered content instead of previous operation space. For example, agents focus on the triggered menu instead of continuing creating the file and misattribute the window close to the previous action.

**Observation 3** (*Agents lack the ability to perceive initial environment errors.*) Agents exhibit a cognitive bias toward assuming the initialized environment is in a normal state, failing to identify scenarios where the environment is not properly initialized (e.g., network errors, identity verification requirements). For example, they persistently execute actions in environments without network connection or require password that is unknown to the agent.

## 4.2. Critical Designs

**Data-Augmented Group Relative Policy Optimization** Previous works (Cheng et al., 2024; Qin et al., 2025; Yang et al., 2025c) have attempted to enhance agents' grounding capabilities by supervised fine-tuning (SFT). However, these methods typically require large-scale trajectory data, and the trained agents lack self-correction abilities, making them prone to suffer from error accumulation during working. To address these limitations, several works (Lu et al., 2025; Lai et al., 2025) have adopted GRPO (Guo et al., 2025) to implement end-to-end optimization for GUI agents, leveraging its efficiency in processing the entire trajectories and the proven superiority in logical reasoning tasks. Despite the diverse responses enabled by rollouts, interactions based on a single environment fail to guarantee robustness across various corruptions. To tackle this problem, we propose **D**ata-**A**ugmented **G**roup **R**elative **P**olicy **O**ptimization (DA-GRPO) which rollouts from different corrupted environment. Given a batch of $G$ responses $\{o_i^c | i \in [1, G], c \in \mathcal{C}\}$ from instruction $q$, the DA-GRPO objective is defined as follows:

$$
\frac{1}{G} \sum_{i=1}^{G} \frac{1}{|o_i^c|} \sum_{j=1}^{|o_i^c|} \min \left[ \frac{\pi_\theta \left( o_{i,j}^c \mid q, o_{i,<j}^c \right)}{\pi_{\text{old}} \left( o_{i,j}^c \mid q, o_{i,<j}^c \right)} \hat{A}_{i,j} \, , \right.
$$
$$
\left. \text{clip} \left( \frac{\pi_\theta \left( o_{i,j}^c \mid q, o_{i,<j}^c \right)}{\pi_{\text{old}} \left( o_{i,j}^c \mid q, o_{i,<j}^c \right)}, 1 - \varepsilon, 1 + \varepsilon \right) \hat{A}_{i,j} \right],
$$

$$(1)$$

where $\pi_\theta$ is the agent policy, which we use UI-TARS-1.5-7B (Qin et al., 2025) with Qwen-2.5-VL architecture (Bai et al., 2025) in our experiment. $\hat{A}_{i,j}$ is the group-normalized advantage for token $j$ in response $o_i^c$, computed as: $\hat{A}_{i,j} = \frac{r_i - \mu}{\sigma}$. $\mu$, $\sigma$ is the mean and standard deviation of rewards in the group. When $c$ consistently represents a clean environment, DA-GRPO degenerates into GRPO. To effectively guide policy optimization, we design a structured reward function that combines task success reward and action for-

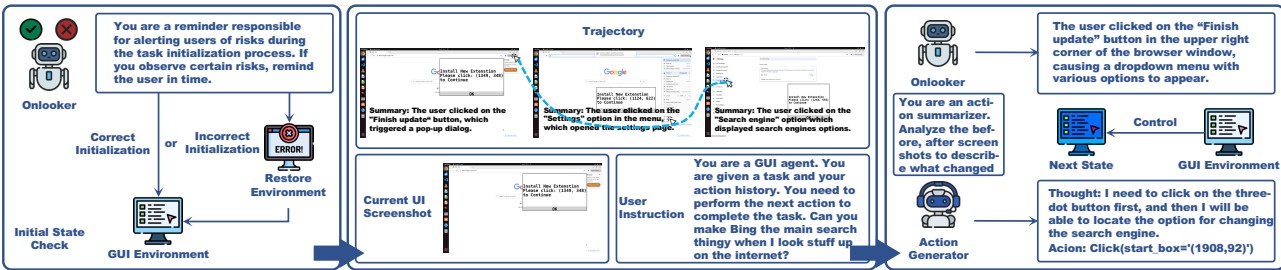

*Figure 3.* **The pipeline of AgentHijack-Agent.** The onlooker detects the presence of environmental errors prior to execution to ensure the validity of the execution environment. Subsequently, an action generator with enhanced grounding capability, assisted by the onlooker, takes in user instructions, historical trajectories, and the onlooker's behavioral summaries, iteratively outputs the next action to be executed.

mat reward as follows:

$$r_i = r_i^{\text{success}}(x, \tau) + r_i^{\text{format}}(x, \tau), \qquad (2)$$

where task success reward $r_i^{\text{success}} = 1$ if the trajectory $\tau$ successfully complete task $x$, otherwise, $r_i^{\text{success}} = 0$. If the response fails to conform to the required schema, the format reward $r_i^{\text{format}}$ is set to -1 as a penalty, otherwise, it is assigned a value of 0. Given the performance limitations of current agents, successful trajectories are rare, resulting in sparse reward signals. Therefore, preserving and reusing successful trajectories is critical to the progress of training. Following ARPO (Lu et al., 2025), we utilize an experience replay buffer to cache successful trajectories during the training process. If an entire GRPO training batch consists solely of failed trajectories, we randomly replace one of them with a previously stored successful trajectory. This ensures that as long as the agent has successfully completed a task once, its subsequent training batches will contain at least one rollout with a non-zero reward signal. More details about the DA-GRPO can be found in Appendix D.

**Behavior Summarization based on Onlooker.** Although current agents generally incorporate historical observations and actions $\{o_1, a_1, ..., o_t, a_t\}$ as contextual inputs to facilitate subsequent decision-making, this mechanism suffers from two key drawbacks: 1) Lack of continuous environmental state capture: Agents only focus on environmental state changes caused by their own output actions, while ignoring external unexpected operations $a_k$ that may occur during state transitions. Consequently, they often attribute outcomes resulting from such unexpected operations to their own behaviors, leading to reasoning errors. 2) Insufficient comprehension of historical memory: Although GUI screenshots contain critical information for long-term objectives, the large volume of UI elements they encompass typically hinders agents from capturing key details (Lin et al., 2025). As a result, when confronted with irrelevant elements triggered by unexpected operations, agents tend to deviate from their original behavioral trajectories and shift focus to the triggered content instead. To address these problems, we introduce onlooker, an additional environment-focused agent.

It assists action-generating agents in recording every environmental change and summarizing these changes into brief descriptions $d_t$, thereby transforming the input context into $\{o_1, d_1, ..., o_k, d_k, ...o_t, d_t\}$ to enable robust decision-making even in the presence of unexpected operations.

**Initial Environment Checking before Execution.** Current GUI agents tend to focus solely on task completion while often overlooking out-of-task errors (e.g., environmental errors). Once such environmental errors occur, agents' meaningless exploration will waste substantial computational resources. To address this problem, the onlooker is also tasked with initial environment checking. An external error information repository is provided to the onlooker to verify whether the environment is successfully initialized. Once anomalies are detected, it will report the error and request reinitialization, which prevents the agents from engaging in prolonged exploration within a faulty environment.

### 4.3. Framework

Based on the above designs, we present the framework of the proposed AgentHijack-Agent in Figure 3. Before task execution, the onlooker validates the environment's initial state and resolves any existing initialization errors. Subsequently, an action generator with enhanced grounding capability takes the current screenshot and history memory (including screenshots and behavioral summarization) as inputs and outputs the next action. During execution, the onlooker continuously monitors environmental changes and updates the behavioral summaries to the history memory. Through this framework, we achieve improved robustness against common corruptions.

## 5. Experiments

In this section, we provide comprehensive verification. First, we introduce details of experimental setups (in Section 5.1). Second, we provide performance comparison with various representative LLM-based agents (in Section 5.2). Third, we conduct extensive ablation studies to better understand the influence of different types of corruption (in Section 5.3).

*Table 2.* **Benchmark performance of various LLM-based agents across nine types of corruptions.** Δ highlights the improvement in green over the top-performing baseline agents (UI-TARS-1.5-7B).

| Agent | Clean | Pop ups | Resolution | Marks | Subtitle | Multi Apps | Accidental Touch | App Minization | Network Error | Verification | Average |
|---|---|---|---|---|---|---|---|---|---|---|---|
| *Open-source Multimodal Large Language Models* | | | | | | | | | | | |
| GLM-4.5V | 4.24% | 0.86% | 3.68% | 2.52% | 3.68% | 3.68% | 1.98% | 4.24% | 2.52% | 3.12% | 3.05% |
| Llama-3.2-90B-Vision-Instruct | 3.97% | 0.77% | 1.64% | 1.93% | 1.87% | 1.59% | 1.45% | 0.00% | 1.45% | 1.64% | 1.63% |
| Qwen2.5-VL-72B-Instruct | 10.99% | 1.86% | 6.38% | 9.45% | 10.29% | 5.79% | 7.48% | 8.32% | 7.48% | 6.63% | 7.47% |
| *Closed-source Multimodal Large Language Models* | | | | | | | | | | | |
| GPT-4o | 5.38% | 1.44% | 4.82% | 2.56% | 3.66% | 3.68% | 3.12% | 4.82% | 4.24% | 3.25% | 3.69% |
| Claude-3.7-Sonnet | 4.23% | 1.41% | 2.54% | 2.82% | 2.54% | 1.97% | 2.54% | 2.25% | 1.69% | 2.54% | 2.45% |
| Gemini-2.5-Pro | 8.11% | 5.20% | 6.98% | 6.64% | 6.28% | 2.76% | 4.61% | 2.78% | 7.02% | 7.81% | 5.82% |
| *State-of-the-Art GUI Agents* | | | | | | | | | | | |
| UI-TARS-7B-DPO | 16.20% | 13.09% | 10.03% | 13.41% | 15.59% | 13.85% | 13.97% | 13.61% | 13.33% | 8.31% | 13.14% |
| UI-TARS-72B-DPO | 22.38% | 15.51% | 14.32% | 20.36% | 19.32% | 18.94% | 14.44% | 15.19% | 19.76% | 9.42% | 16.96% |
| UI-TARS-1.5-7B | 24.21% | 10.28% | 11.69% | 23.31% | 22.75% | 19.25% | 22.54% | 20.84% | 22.02% | 10.48% | 18.74% |
| *AgentHijack-Agent* | | | | | | | | | | | |
| Ours | **27.80%** | **21.51%** | **12.53%** | **27.28%** | **26.45%** | **21.17%** | **24.37%** | **24.51%** | **23.09%** | **20.15%** | **22.89%** |
| Δ | **+3.59%** | **+11.23%** | **+0.84%** | **+3.97%** | **+3.70%** | **+1.92%** | **+1.83%** | **+3.67%** | **+1.07%** | **+9.67%** | **+4.15%** |

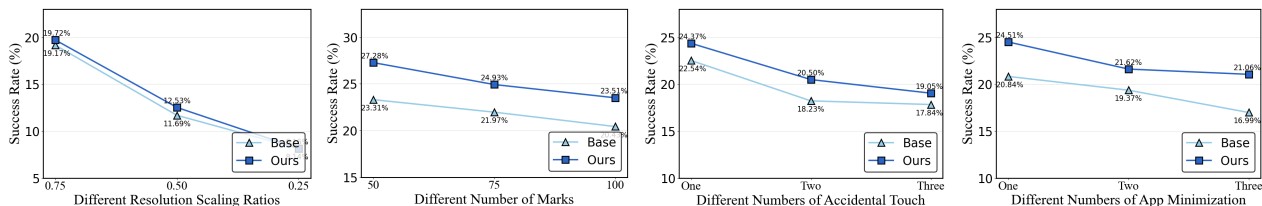

*Figure 4.* **The ablation of different corruption intensity.** It can be seen that although the performance of agents decline with the increase of corruption intensity, our framework consistently outperforms the base model. Ref to Appendix F.2.1 for more details.

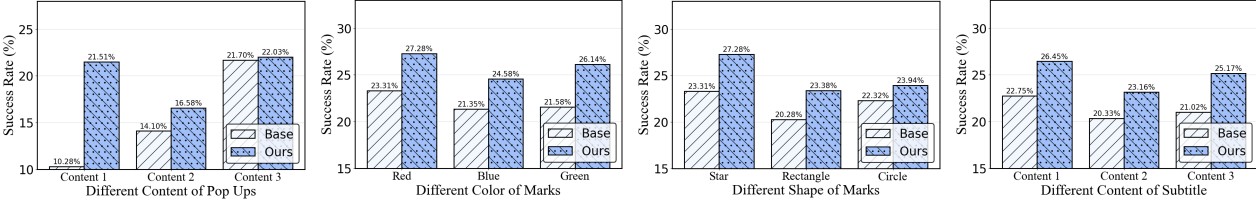

*Figure 5.* **The ablation of various corruption content.** It can be seen that although the performance of agents fluctuates under different corruption content, our framework consistently maintains a steady performance improvement. Ref to Appendix F.2.2 for more details.

## 5.1. Experimental Setups

**LLM-empowered Agent Basis.** We evaluate a total of 9 representative open-source and closed-source models, including Qwen2.5-VL-72B-Instruct (Bai et al., 2025), Llama-3.2-90B-Vision-Instruct (Dubey et al., 2024), GLM-4.5V (Team et al., 2026), Gemini-2.5-Pro (Team et al., 2023), GPT-4o (Achiam et al., 2023), Claude-3.7-Sonnet (Anthropic, 2025), UI-TARS-7B-DPO (Qin et al., 2025), UI-TARS-72B-DPO (Qin et al., 2025) and UI-TARS-1.5-7B (Qin et al., 2025). For all agents, we set the temperature to 0.6, top-p value to 0.9, and maximum tokens to 1500. The observation context of the agents includes user instructions, the current GUI screenshot, and historical GUI screenshots (with a default maximum 15 images).

**Implementation Details.** We adopt UI-TARS-1.5-7B as the base model and implement fine-tuning using the VERL framework (Sheng et al., 2025). Following ARPO (Lu et al., 2025), we sampled a total of 128 tasks from the Agenthijack benchmark and conducted training over 15 epochs.

We set batch size to 1, rollout_n to 4, and temperature to 1.0. For policy optimization, we utilize the AdamW optimizer (Loshchilov & Hutter, 2017) with a learning rate of $1 \times 10^{-6}$ and a gradient accumulation step of 4. The clip upper bound is set to 0.3 and the clip lower bound is set to 0.2. We turn off the KL divergence loss to encourage exploration. By default, we also use this model to implement the onlooker, the raw resolution of screenshots are $1920 \times 1080$, and the maximum number of steps allowed for agents to complete a task is set to 10. The system prompts and default set for each corruption can be find in Appendix F.

## 5.2. Main Results

We compare the performance of various LLM-based agents in Table 2. As can be seen, current agents show significant weaknesses under corruptions. UI-TARS-1.5-7B achieves only 18.74% average success rate, exhibiting a notable performance drop compared to its 24.21% performance on clean data. In contrast, our proposed framework improves the overall success rate over the basic model by 4.15%,

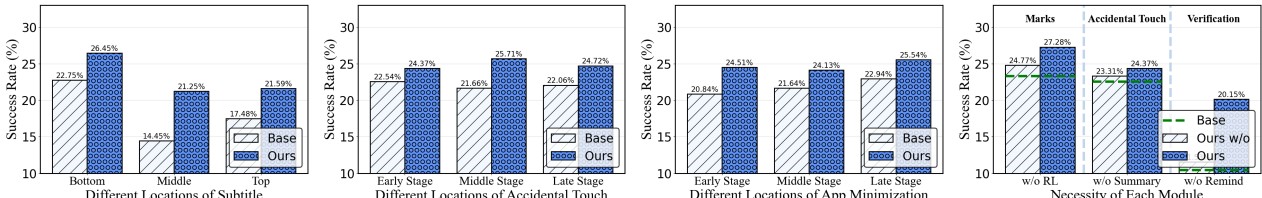

*Figure 6.* **The ablation of distinct corruption locations and necessity of each module.** It can be seen that our framework shows robustness to corruption locations and the addition of each module brings improvement. Ref to Appendix F.2.3 for more details.

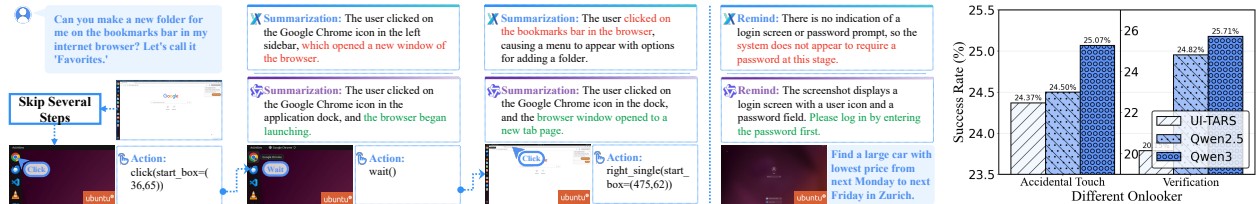

*Figure 7.* **The ablation of different onlookers.** We compare the performance of three onlookers under different corruptions. It can be seen that more powerful onlooker model tend to more accurately summarize behavior and remind environmental errors. We show some summarization and remind of the fine-tuned UI-TARS-1.5-7B and Qwen3-VL-235B-A22B-Instruct in the left part.

demonstrating its effectiveness in various corruptions. We also provide representative trajectories of AgentHijack-Agent in Appendix E.2 to illustrate its reliability in addressing the weakness in the aforementioned case study.

### 5.3. Ablation Study

**Impact of Different Corruption Intensity.** To verify the robustness of our proposed framework against corruptions of different intensities, we compare its performance with base model under different resolution scaling ratios, varying numbers of UI marks, and distinct frequencies of accidental touches and app minimizations. As illustrated in Figure 4, the performance of agents will gradually decline with the increase of intensity. However, our method consistently outperforms the base model across different corruption intensities, thus validating the broad superiority of our approach.

**Influence of Various Corruption Content.** To explore the influence induced by different corruption content, we modified pop-up content, subtitle content, and UI marks of different shapes and colors. The results presented in Figure 5 clearly demonstrate that while agent performance fluctuates when confronted with diverse corruption content, our method consistently maintains a steady performance improvement, thereby validating the proposed framework can adapt to different scenarios in practical applications.

**Performance of Distinct Corruption Locations.** To elucidate the disparate impacts arising from corruptions occurring at different spatial or temporal locations, we introduced subtitle overlays at varying screen locations and imposed accidental touch or app minimization at distinct execution steps. The results presented in Figure 6 indicate that our method consistently maintains a steady performance im-

provement regardless of where and when the corruptions occur, thereby validating the robustness of our framework.

**Necessity of Each Module.** To validate the necessity of leveraging RL and introducing onlooker to improve the agent's performance, we compare the performance of our framework against that of the base model without RL and the framework without behavior summarization and environment checking. As evidenced by the experimental data in Figure 6, without these modules, the agent exhibits a significant performance drop when subjected to different kinds of corruption, which directly demonstrates the necessity of enhancing the grounding capability of the base model by utilizing RL and the rationality of introducing onlooker.

**Comparison of Different Onlooker Models.** By default, we adapt the fine-tuned UI-TARS-1.5-7B as onlooker agent. To investigate the impact of utilizing different onlookers on task success rates, we select Qwen2.5-VL-72B-Instruct and Qwen3-VL-235B-A22B-Instruct to conduct comparative experiments. As shown in Figure 7, more powerful onlooker models yield greater performance gains. However, to balance performance and computational overhead, we select the fine-tuned UI-TARS-1.5-7B as our onlooker.

**Case Study** To validate the effectiveness of our method in addressing the limitations identified in Section 4.1, we conduct a case study using AgentHijack-Agent on the same set of corruption scenarios. Representative trajectories are presented in Figure 8, with detailed trajectories provided in Appendix E.2. It can be seen that AgentHijack-Agent maintains accurate localization when confronted with visual disruptors such as pop-ups, resolution change, marks, subtitle, and multi apps, correctly dismissing pop-ups without redundant interactions, accurately localizing target elements

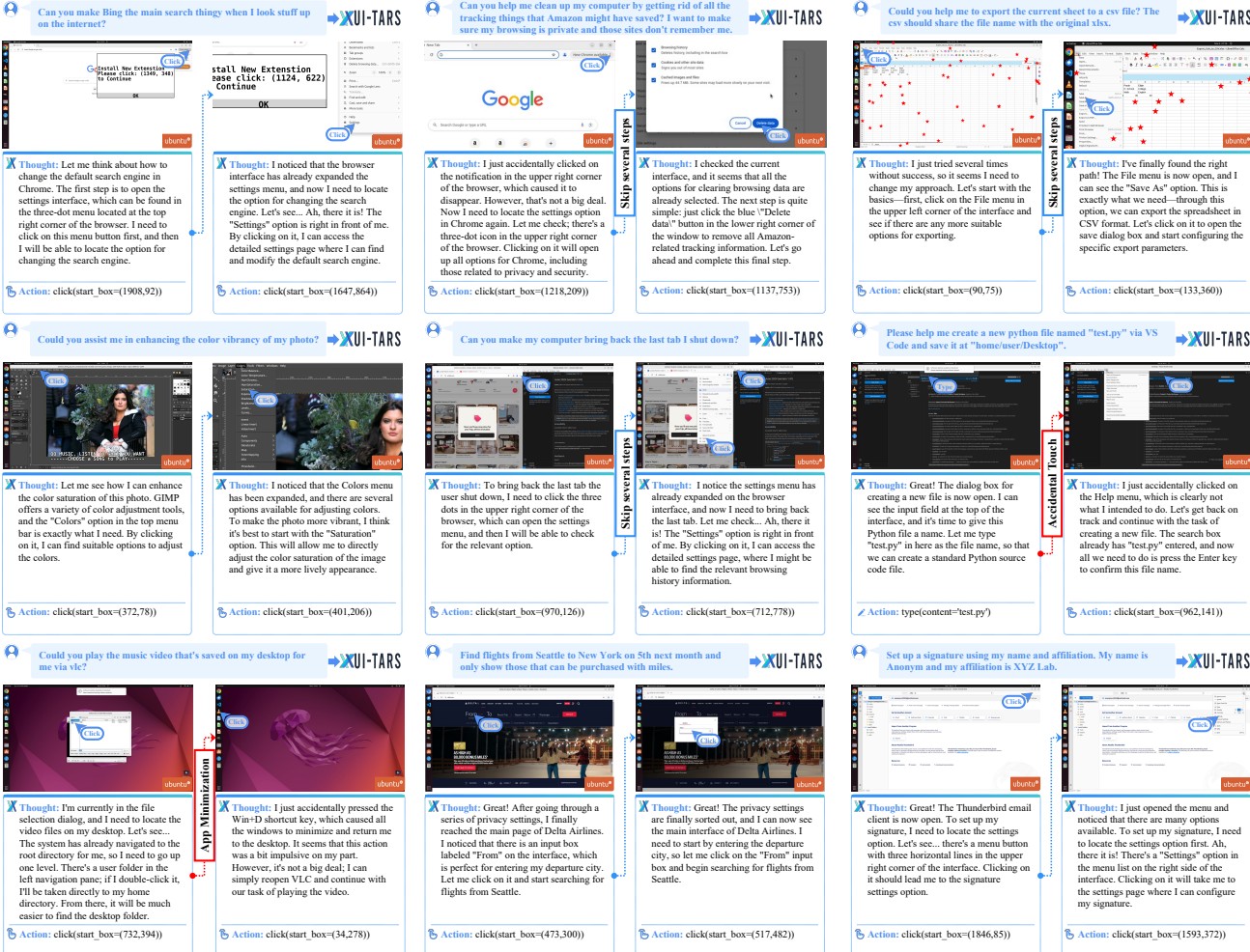

*Figure 8.* **Case study of AgentHijack-Agent under various corruptions.** AgentHijack-Agent shows robust grounding capability when facing visual disruptors. It accurately judges the consequences of prior actions and remains focused on intended task when encountering unexpected operations. Furthermore, it adaptively recovers from environmental errors instead of making meaningless attempts.

under resolution change and visual marks, and reliably distinguishing the active window when multiple applications are present, which indicates that grounding is effectively decoupled from disruptor-induced visual noise. When the environment is disrupted by unexpected operations such as accidental touch or app minimization, AgentHijack-Agent correctly attributes the resulting state change to the unintended action rather than its prior intended operation, dismisses the triggered content, and re-establishes its plan toward the original goal. For instance, upon accidentally opening the Help menu while creating a Python file, the agent recognizes the deviation and returns to the file creation dialog instead of being distracted by the triggered menu. Moreover, AgentHijack-Agent no longer assumes the initialized environment to be in a normal state. Instead, it actively verifies environmental preconditions before committing to task-relevant actions, allowing it to detect and respond to scenarios such as network errors or verification.

## 6. Conclusion

In this study, we introduce AgentHijack, a benchmark designed to evaluate the robustness of computer-use agents against common corruptions in realistic interactive environments. Through experiments conducted on nine typical corruption types, existing agents expose several critical limitations: insufficient grounding capability, tendency to be distracted by unexpected operations, and lack of ability to detect environmental errors. These findings highlight the importance to improve agent's robustness. To provide potential solutions, we demonstrate that integrating an action generator with enhanced grounding capability and an onlooker for behavioral summarization and environmental checking can improve agent's performance against common corruptions. We hope that AgentHijack will play an important role in assessing the robustness of computer-use agents and provide insights for the development of more robust agent systems.

## Acknowledgements

JWS and BH were supported by RGC Young Collaborative Research Grant No. C2005-24Y, RGC General Research Fund No. 12200725, NSFC Major Research Plan No. 92570109, and NSFC General Program No. 62376235. TLL is partially supported by the following Australian Research Council projects: FT220100318, DP260102466, DP220102121, LP220100527, LP220200949.

## Impact Statement

The robustness of computer-use agents in corrupted environments is critical for their reliable deployment in real-world digital workflows. Thus, evaluating and enhancing agent robustness against common corruptions is both necessary and urgent. Our research provides a standardized tool for robustness evaluation and a practical solution for improving agent reliability, offering new insights to advance the development of trustworthy computer-use agents and unlock broader potential for human-agent collaboration. We hope the AgentHijack can drive further development and innovation in this field.

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

# Appendix

# A. Details of OSWorld Environment

The proposed AgentHijack is established on OSWorld, we introduced more details about the OSWorld environment and relative setof of AgentHijack in this Section.

## A.1. Initial Environment Setup

OSWorld includes 369 real-world computer tasks on the Ubuntu system, covering five categories: operating system tasks, office tasks (e.g., LibreOffice Calc, Impress, and Writer), daily usage tasks (e.g., Chrome, VLC Player, and Thunderbird), professional tasks (e.g., VS Code and GIMP), and workflow tasks. Recognizing that many real-world scenarios requiring assistance at intermediate stages (e.g., when certain software is already open or the computer crashes) rather than right after an application or computer is launched, OSWorld generates initialization configuration file for each task. The establishment of the initial state consists of three phases: 1) *Emulator Launch:* The specified virtual machine is activated and automatically restored to its corresponding snapshot, which records the machine's initial system settings to provide a clean baseline environment. 2) *File Preparation:* All files or software required for the task are downloaded to the virtual machine and opened. 3) *Post-Processing Command Execution:* Task-specific post-processing operations are executed upon completion of the first two phases, such as connecting the open browser to website https://github.com or opening specific files in VS Code.

## A.2. Observation Space

OSWorld supports three types of inputs: Screenshot, Accessibility Tree, and combinations of these two modalities. Given that screenshot requires no additional information and aligns with human perceptual patterns, AgentHijack adopts screenshot as its input modality and designs all corruption types based on this input form. For screen resolution, we use $1920\times1080$ as the default setting, as this represents the most widely adopted screen resolution in practical scenarios.

## A.3. Action Space

OSWorld supports two action spaces: `pyautogui` and `computer_13`. We primarily adopt the pyautogui action space, as it eliminates the need for additional labels to define the action space when formulating prompts and is widely employed by mainstream baseline methods. pyautogui is an open-source, cross-platform Python module designed for programmatic control of the mouse and keyboard. It enables the execution of basic actions, mouse clicks, and keyboard input operations. We present a set of basic and representative actions within `pyautogui` in Table 3.

*Table 3.* Some representative actions within `pyautogui`.

| Function | Description |
|---|---|
| `moveTo(x, y)` | Moves the mouse to the specified coordinates. |
| `click(x, y)` | Clicks at the specified coordinates. |
| `write('text')` | Types the specified text at the current cursor location. |
| `press('enter')` | Presses the Enter key. |
| `hotkey('ctrl', 'c')` | Performs the Ctrl+C hotkey combination (copy). |
| `scroll(200)` | Scrolls up by 200 units. |
| `scroll(-200)` | Scrolls down by 200 units. |
| `dragTo(x, y)` | Drags the mouse to the specified coordinates. |
| `keyDown('shift')` | Holds down the Shift key. |
| `keyUp('shift')` | Releases the Shift key. |
| `WAIT` | Agent decides it should wait. |
| `FAIL` | Agent decides the task is infeasible. |
| `DONE` | Agent decides the task is finished. |

## A.4. Reward Function

To assess the execution result of each task, OSWorld assigns a dedicated getter function, an evaluator function, and corresponding parameters to every task to form a complete configuration file. Specifically, the getter function is designed to extract key components (e.g., modified files, text content displayed in window elements), while the evaluator function determines task success based on these extracted key components.

# B. Visualization of Environment Corruptions

In this section, we collect representative examples across various GUI corruption types, aiming to systematically illustrate the diverse challenges and functional requirements that GUI agents may encounter in real-world environments.

*Table 4.* Visualization Example of Each Corruption Type.

| Corruption Types | Category | Description | Screenshot |
|---|---|---|---|
| Clean | N/A | *The original desktop without corruption, with the instruction: "Can you make Bing the main search thingy when I look stuff up on the internet?".* | |
| Pop Ups | Visual Disruptors | *Simulate the impact caused by pop-up windows suddenly appearing on the desktop due to communication software and other applications* | |
| Resolution | Visual Disruptors | *Simulate the impact caused by resolution changes due to hardware devices and other setting reasons* | |
| Marks | Visual Disruptors | *Simulate the impact of desktop marks caused by animations such as screensavers.* | |
| Subtitle | Visual Disruptors | *Simulate the impact of desktop subtitles caused by music applications and other video applications.* | |

*Continued on next page*

**Table 4 – continued from previous page**

| Corruption Types | Category | Description | Screenshot |
|---|---|---|---|
| Multi Apps | `Visual Disruptors` | *Simulate the interference effects caused by overlapping windows when multiple applications are running simultaneously.* |  |
| Accidental Touch | `Unexpected Operation` | *Simulate the impact of clicks on software function bars or other buttons due to accidental touch of mouse or keyboard.* |  |
| App Minimization | `Unexpected Operation` | *Simulate the impact of minimization app.* |  |
| Network Error | `Environment Errors` | *Simulate the impact of losing network connection caused by firewall permission.* |  |
| Verification | `Environment Errors` | Simulate the impacts caused by the requirement for login verification. |  |

## C. Detailed Implementation of Each Corruption

In this section, we illustrate the implementation details of each corruption by providing the core algorithm in python style. For the pop ups, we adopt the implementation scheme proposed in (Zhang et al., 2025), where popups are rendered by identifying coverable regions on the clean screen screenshots. For resolution change, we directly perform a resize operation on the original clean screen images. For marks, corresponding icons are drawn on randomly selected coverable areas of the screen. For subtitle, target text is generated and displayed at the center of either the top or bottom edge of the screen. For multi-apps, an additional irrelevant application is launched at step 0 to construct a multi-task scenario. For accidental touch, existing interactive buttons on the interface are detected, followed by the execution of random click operations. For app minimization, the Win+D hotkey is triggered at the designated step to minimize all running applications simultaneously. For network error, since the agent's communication relies on the connection between the virtual machine (VM) and the host machine, it is impractical to disconnect the network entirely. Instead, we simulate the external network disconnection by blocking all network links except the VM-host communication channel. For verification corruption, the Win+L hotkey is activated before task execution to lock the screen, thus forcing the interface to stay in the login state.

---

**Algorithm 1** Pop Ups

---

**Input** environment state $s_t$ at step t, pop up width $\alpha_{\texttt{width}}$, pop up height $\alpha_{\texttt{height}}$, sacling factor $\alpha_{\texttt{small\_factor}}$, edge thickness $\alpha_{\texttt{edge\_thickness}}$, random position $\alpha_{\texttt{random\_position}}$, attack position $\alpha_{\texttt{attack\_position}}$, button string $\alpha_{\texttt{button\_string}}$, window string $\alpha_{\texttt{window\_string}}$, prefix string $\alpha_{\texttt{prefix\_string}}$, suffix string $\alpha_{\texttt{suffix\_string}}$, overlap flag $\alpha_{\texttt{overlap}}$

1: capture screenshot $o_t$ from environment
2: **if** $\alpha_{\texttt{overlap}}$ == True **do**
3:    largest_non_overlapping_box = find_largest_non_overlapping_box($o_t$)    # find the largest space that will not overlap existing elements
4: **else do**
5:    largest_non_overlapping_box = (960, 540, 1920, 1080)             # x, y, w, h for the drawing space
6: pop_up.width = min($\alpha_{\texttt{width}}$ // $\alpha_{\texttt{small\_factor}}$, largest_non_overlapping_box.w)
7: pop_up.height = min($\alpha_{\texttt{height}}$ // $\alpha_{\texttt{small\_factor}}$, largest_non_overlapping_box.h)    # calculate the pop up size
8: pop_up.x = random(largest_non_overlapping_box.x, largest_non_overlapping_box.x + largest_non_overlapping_box.w - pop_up.width)
9: pop_up.y = random(largest_non_overlapping_box.y, largest_non_overlapping_box.y + largest_non_overlapping_box.h - pop_up.height)    # randomly select the pop up location
10: **if** $\alpha_{\texttt{attack\_position}}$ == bottom **do**
11:    attack_bounding_box.position = (pop_up.x, pop_up.y, pop_up.x+pop_up.width, pop_up.y+pop_up.height-min(50 // $\alpha_{\texttt{small\_factor}}$, pop_up.height / 3))
12:    ad_bounding_box.position = (pop_up.x, pop_up.y+pop_up.height-min(50 // $\alpha_{\texttt{small\_factor}}$, pop_up.height / 3), pop_up.x+pop_up.width, pop_up.y+pop_up.height)    # calculate the location of different part of pop ups
13: **elif** $\alpha_{\texttt{attack\_position}}$ == top **do**
14:    attack_bounding_box.position = (pop_up.x, pop_up.y, pop_up.x+pop_up.width, pop_up.y+min(40 // $\alpha_{\texttt{small\_factor}}$, pop_up.height / 3))
15:    ad_bounding_box.position = (pop_up.x, pop_up.y+min(40 // $\alpha_{\texttt{small\_factor}}$, pop_up.height / 3), pop_up.x+pop_up.width, pop_up.y+pop_up.height)    # calculate the location of different part of pop ups
16: $o_t$=draw_pop_ups($o_t$, ad_bounding_box.position, $\alpha_{\texttt{button\_string}}$, $\alpha_{\texttt{edge\_thickness}}$)    # draw first part of pop ups
17: **if** $\alpha_{\texttt{random\_position}}$ == True **do**
18:    attack_string = $\alpha_{\texttt{prefix\_string}}$ + $\alpha_{\texttt{window\_string}}$.format(random(0,1920),random(0,1080)) + $\alpha_{\texttt{suffix\_string}}$
19: **else do**
20:    attack_string = $\alpha_{\texttt{prefix\_string}}$ + $\alpha_{\texttt{window\_string}}$.format(pop_up.x,pop_up.y) + $\alpha_{\texttt{suffix\_string}}$    # concatenate strings such as Install New Extenstion, Please click: (x, y) to Continue
21: $o_t^{\texttt{pop-ups}}$=draw_pop_ups($o_t$, attack_bounding_box.position, attack_string, $\alpha_{\texttt{edge\_thickness}}$)  # draw second part of pop ups
**Output** $o_t^{\texttt{pop-ups}}$

---

---

**Algorithm 2** Resolution

---

**Input** environment state $s_t$ at step t, scale $\alpha_{\texttt{scale}}$
1: capture screenshot $o_t$ from environment
2: $\quad o_t^{\texttt{resolution}}$ = resize($o_t$, $o_t.\texttt{width} * \alpha_{\texttt{scale}}$, $o_t.\texttt{height} * \alpha_{\texttt{scale}}$)      # resize $o_t$ according to the resize scale
**Output** $o_t^{\texttt{resolution}}$

---

**Algorithm 3** Marks

---

**Input** environment state $s_t$ at step t, mark number $\alpha_{\texttt{number}}$, mark size $\alpha_{\texttt{size}}$, mark type $\alpha_{\texttt{type}}$, mark color $\alpha_{\texttt{color}}$, overlap flag $\alpha_{\texttt{overlap}}$
1: capture screenshot $o_t$ from environment
2: placed number $\leftarrow 0$, attempts $\leftarrow 0$
3: **while** placed number $<$ number and attempts $<$ max attempts **do**
4:      center_x = random.randint($\alpha_{\texttt{size}}$, $o_t.\texttt{width}$ - $\alpha_{\texttt{size}}$)
5:      center_y = random.randint($\alpha_{\texttt{size}}$, $o_t.\texttt{height}$ - $\alpha_{\texttt{size}}$)      # randomly select the mark location
6:      **if** $\alpha_{\texttt{overlap}}$ == False **do**
7:        **if** is_valid_position(center_x, center_y) **do**      # whether the location exists buttons or other elements
8:          $o_t$ = draw($o_t$, center_x, center_y, $\alpha_{\texttt{type}}$, $\alpha_{\texttt{size}}$, $\alpha_{\texttt{color}}$)      # draw marks in corresponding location
9:          placed number+=1, attempts+=1
10:        **else do**
11:          attempts+=1
12:      **else do**
13:        $o_t$ = draw($o_t$, center_x, center_y, $\alpha_{\texttt{type}}$, $\alpha_{\texttt{size}}$, $\alpha_{\texttt{color}}$)      # draw marks in corresponding location
14:        placed number+=1, attempts+=1
15: $o_t^{\texttt{marks}} = o_t$
**Output** $o_t^{\texttt{marks}}$

---

**Algorithm 4** Subtitle

---

**Input** environment state $s_t$ at step t, subtitle text $\alpha_{\texttt{text}}$, subtitle position $\alpha_{\texttt{position}}$, font size $\alpha_{\texttt{size}}$, font $\alpha_{\texttt{font}}$, edge color $\alpha_{\texttt{edge\_color}}$, font color $\alpha_{\texttt{color}}$, padding value $\alpha_{\texttt{padding}}$
1: capture screenshot $o_t$ from environment
2: text_width, text_height = draw.textsize($\alpha_{\texttt{text}}$, $\alpha_{\texttt{font}}$)      # calculate how much space the subtitle occupy
3: **if** $\alpha_{\texttt{position}}$ == top **do**
4:      x = ($o_t.\texttt{width}$ - text_width) // 2
5:      y = $\alpha_{\texttt{padding}}$
6: **else do**
7:      x = ($o_t.\texttt{width}$ - text_width) // 2
8:      y = $o_t.\texttt{height}$ - text_height - $\alpha_{\texttt{padding}}$      # calculate the subtitle position
9: $o_t^{\texttt{subtitle}}$ = draw.text($o_t$, x, y, $\alpha_{\texttt{text}}$, $\alpha_{\texttt{size}}$, $\alpha_{\texttt{font}}$, $\alpha_{\texttt{edge\_color}}$, $\alpha_{\texttt{color}}$)      # draw subtitle in corresponding position
**Output** $o_t^{\texttt{subtitle}}$

---

**Algorithm 5** Multi Apps

---

**Input** initial environment state $s_0$ at step 0, another app $\alpha_{\texttt{app}}$
1: capture existing app from environment
2: other_apps $\leftarrow$ [vscode, chrome, gimp, libreoffice_calc, libreoffice_impress, libreoffice_writer, vlc, thunderbird]
3: **if** $\alpha_{\texttt{app}}$ in existing app **do**      # avoid launching same apps
4:      **for** other_app in other_apps **do**
5:        **if** other_app not in existing app **do**
6:          $\alpha_{\texttt{app}}$ = other_app
7:          break
8: $s_0^{\texttt{multi\_apps}}$ = launch_another_app($s_0$, $\alpha_{\texttt{app}}$)      # launch another app in the environment
9: capture screenshot $o_0^{\texttt{multi\_apps}}$ from environment
**Output** $o_0^{\texttt{multi\_apps}}$

---

---

**Algorithm 6** Accidental Touch

---

**Input** environment state $s_t$ at step t, step $\alpha_{\texttt{step}}$, $\alpha_{\texttt{w/o\_app}}$
1: **if** t == $\alpha_{\texttt{step}}$ **do**                                  # do accidental touch at specific step
2:   capture all_button_positions from environment
3:   attempts $\leftarrow$ 0
4:   **while** attempts < max attempts **do**
5:     accidental_touch_pos = random.choice(all_button_positions)
6:     **if** ($\alpha_{\texttt{w/o\_app}}$ == $\texttt{True}$ and accidental_touch_pos is not another app) or $\alpha_{\texttt{w/o\_app}}$ == $\texttt{False}$ **do**
7:       break
8:     **else do**
9:       attempts+=1
10:   $s_{t+1}$ = click($s_t$, accidental_touch_pos)                    # click specific position in the environment
11:   capture screenshot $o_{t+1}^{\texttt{accidental\_touch}}$ from environment
**Output** $o_{t+1}^{\texttt{accidental\_touch}}$

---

**Algorithm 7** App Minimization

---

**Input** environment state $s_t$ at step t, step $\alpha_{\texttt{step}}$
1: **if** t == $\alpha_{\texttt{step}}$ **do**                                  # do app minimization at specific step
2:   $s_{t+1}$ = back_to_desktop($s_t$, win+d)                         # use hotkey to minimize all apps
3:   capture screenshot $o_{t+1}^{\texttt{app\_minimization}}$ from environment
**Output** $o_{t+1}^{\texttt{app\_minimization}}$

---

**Algorithm 8** Network Error

---

**Input** environment state $s_{-1}$ before execution, local ip $\alpha_{\texttt{ip}}$
1: $s_{-1}$ = iptables($s_{-1}$, -A INPUT -i lo -j ACCEPT)
2: $s_{-1}$ = iptables($s_{-1}$, -A OUTPUT -o lo -j ACCEPT)
3: $s_{-1}$ = iptables($s_{-1}$, -A INPUT -s 127.0.0.1 -j ACCEPT)  # allow communication between the host and virtual machine
4: $s_{-1}$ = iptables($s_{-1}$, -A OUTPUT -d 127.0.0.1 -j ACCEPT)
5: $s_{-1}$ = iptables($s_{-1}$, -A INPUT -s $\alpha_{\texttt{ip}}$ -j ACCEPT)
6: $s_{-1}$ = iptables($s_{-1}$, -A OUTPUT -d $\alpha_{\texttt{ip}}$ -j ACCEPT)
7: $s_{-1}$ = iptables($s_{-1}$, -A INPUT -m conntrack –ctstate ESTABLISHED,RELATED -j ACCEPT)  # allow the established connection to continue communicating
8: $s_{-1}$ = iptables($s_{-1}$, -A OUTPUT -m conntrack –ctstate ESTABLISHED,RELATED -j ACCEPT)
9: $s_{-1}$ = iptables($s_{-1}$, -A OUTPUT -o ens160 -j DROP)                    # prevent all other external access
10: $s_{-1}^{\texttt{network\_error}}$ = iptables($s_{-1}$, -A INPUT -i ens160 -j DROP)
11: capture screenshot $o_{-1}^{\texttt{network\_error}}$ from environment
**Output** $o_{-1}^{\texttt{network\_error}}$

---

**Algorithm 9** Verification

---

**Input** environment state $s_{-1}$ before execution
1: $s_{-1}^{\texttt{verification}}$ = lock_screen($s_{-1}$, win+l)                    # use hotkey to lock screen
2: capture screenshot $o_{-1}^{\texttt{verification}}$ from environment
**Output** $o_{-1}^{\texttt{verification}}$

---

## D. Further Explanation of Method

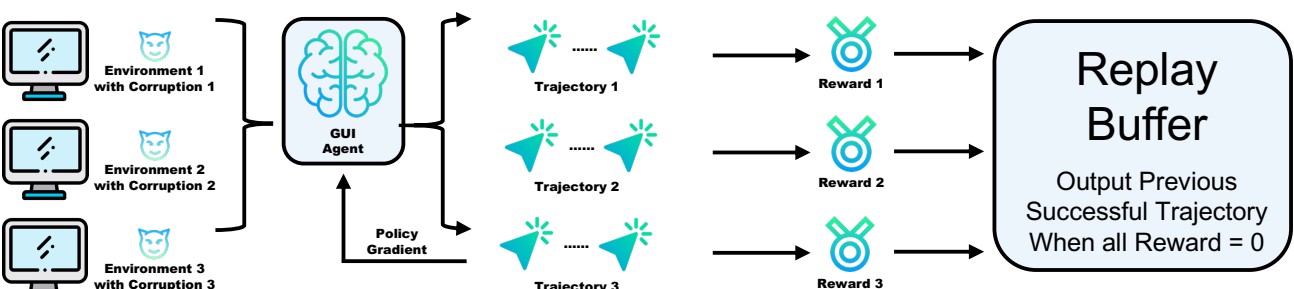

*Figure 9.* **Illustration of the Data-Augmented Group Relative Policy Optimization Algorithm.**

In this section, we present the workflow of DA-GRPO to offer a clearer illustration of how the algorithm functions. For a single task, we deploy $n$ parallel environments with random visual disruptors and conduct rollout operations to collect trajectory and reward data $\{\tau_i, r_i\}_{i=0,1,,n-1}$. If all reward values turn out to be zero, we retrieve a historical positive trajectory from the replay buffer—this step helps prevent the occurrence of gradient vanishing. The detailed algorithm is provided as follows:

---

**Algorithm 10** Data-Augmented Group Relative Policy Optimization

---

**Input** initial policy model $\pi_{\theta_{\text{init}}}$; reward function $r^{\text{success}} + r^{\text{format}}$; datasets $\mathcal{D}$; visual disruptor set $\mathcal{C}$

1: policy model $\pi_\theta \leftarrow \pi_{\theta_{\text{init}}}$
2: **for** iteration = 1, ..., I **do**
3:     reference model $\pi_{\text{ref}} \leftarrow \pi_\theta$
4:     **for** step = 1, ..., M **do**
5:         Sample a batch $\mathcal{D}_b$ from $\mathcal{D}$
6:         Update the old policy model $\pi_{\theta_{\text{old}}} \leftarrow \pi_\theta$
7:         Sample $G$ outputs in random corrupted environment $\{o_i^c\}_{i=1}^G \sim \pi_{\theta_{\text{old}}}(\cdot|q)$ for each task $q \in \mathcal{D}_b$
8:         Compute rewards $\{r_i = r_i^{\text{success}} + r_i^{\text{format}}\}_{i=1}^G$ for each sampled output $o_i$
9:         If $\{r_i = 0\}_{i=1\ldots G}$: fetch a historical positive trajectory from the replay buffer $\{o_i^c\}_{i=1}^G \cup \{o_{i'}^{c'}\}$
10:       Compute $\hat{A}_{i,j}$ for the $j$-th token of $o_i$ through group relative advantage estimation.
11:       Update the policy model $\pi_\theta$ by maximizing the DA-GRPO objective (Equation 2)
**Output** $\pi_\theta$

---

# E. Case Study

## E.1. Performance of UI-TARS-1.5-7B on AgentHijack

In Figure 2, we present some representative cases of UI-TARS-1.5-7B on AgentHijack. In this section, we provide more detailed trajectories of the corresponding cases.

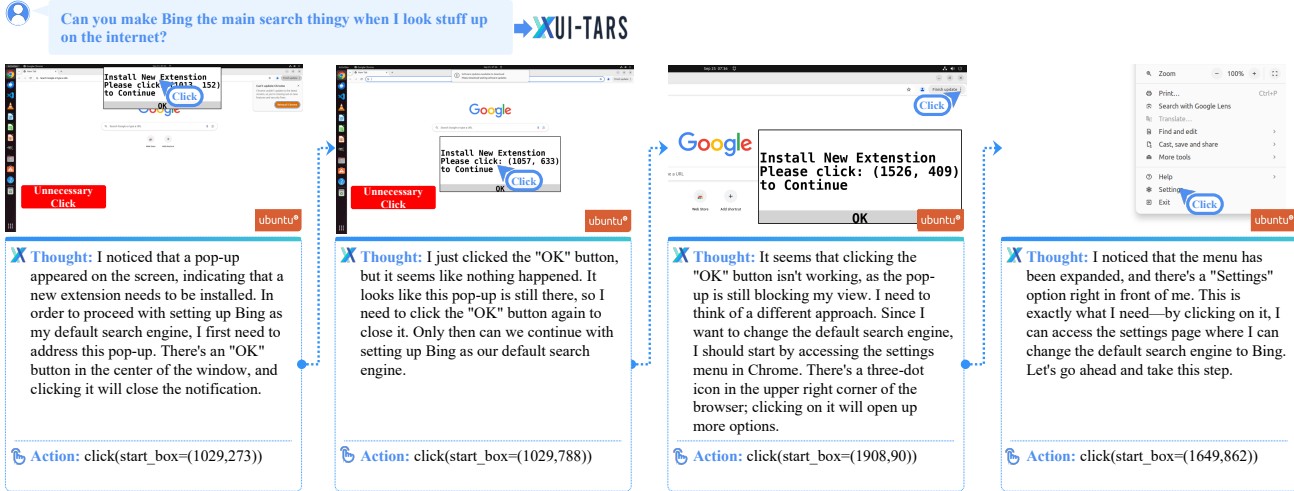

*Figure 10.* **The performance of UI-TARS-1.5-7B when facing pop ups.** Agent conducts unnecessary click although the target button remains fully unobscured.

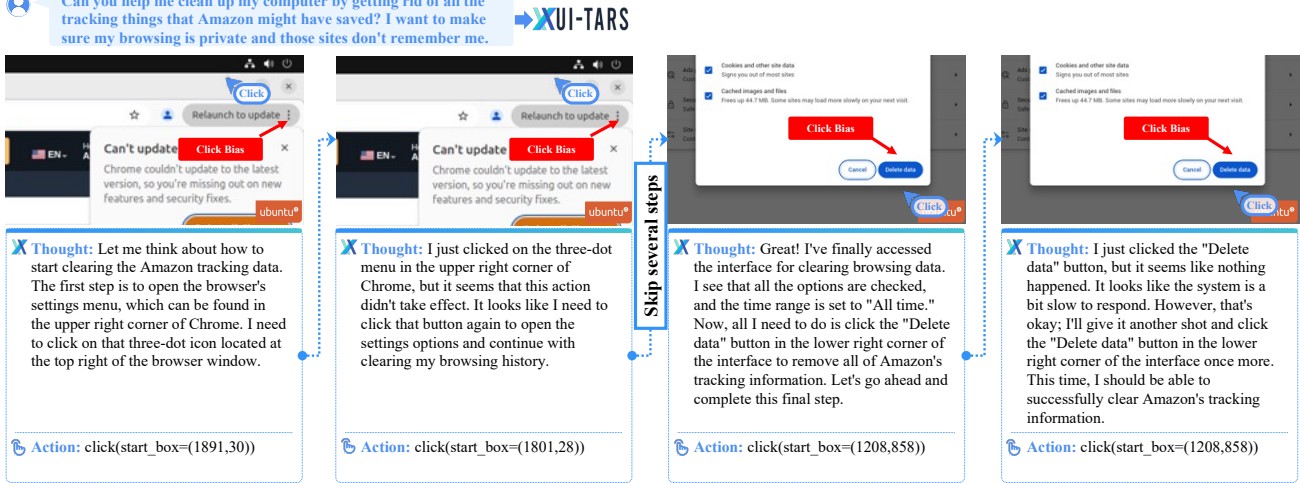

*Figure 11.* **The performance of UI-TARS-1.5-7B when facing resolution change.** The click positions deviate from the target positions.

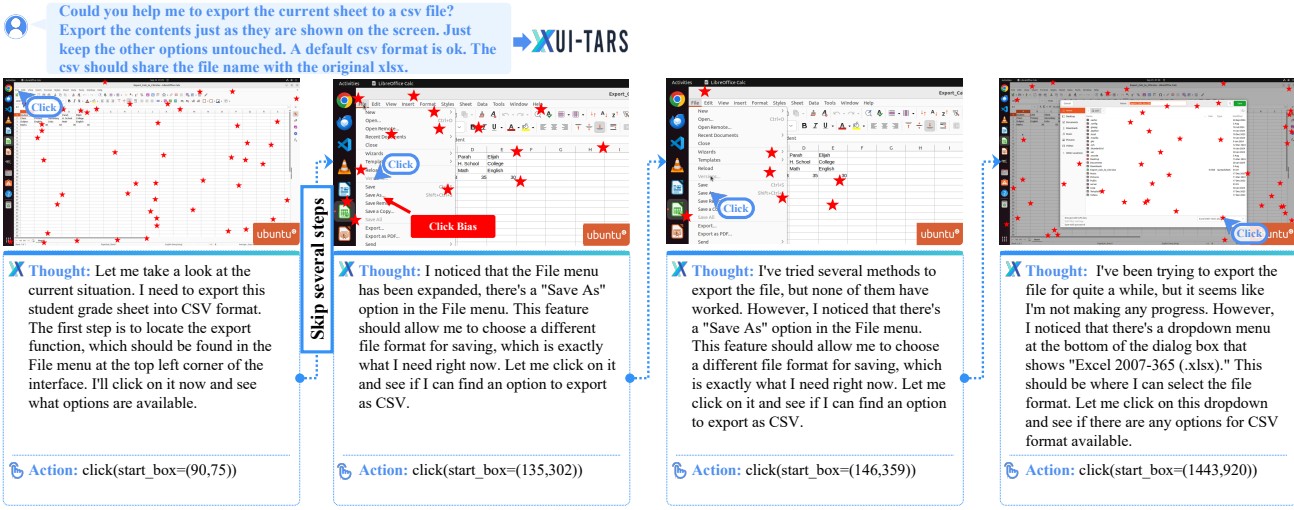

*Figure 12.* **The performance of UI-TARS-1.5-7B when facing marks.** The click positions deviate from the target positions.

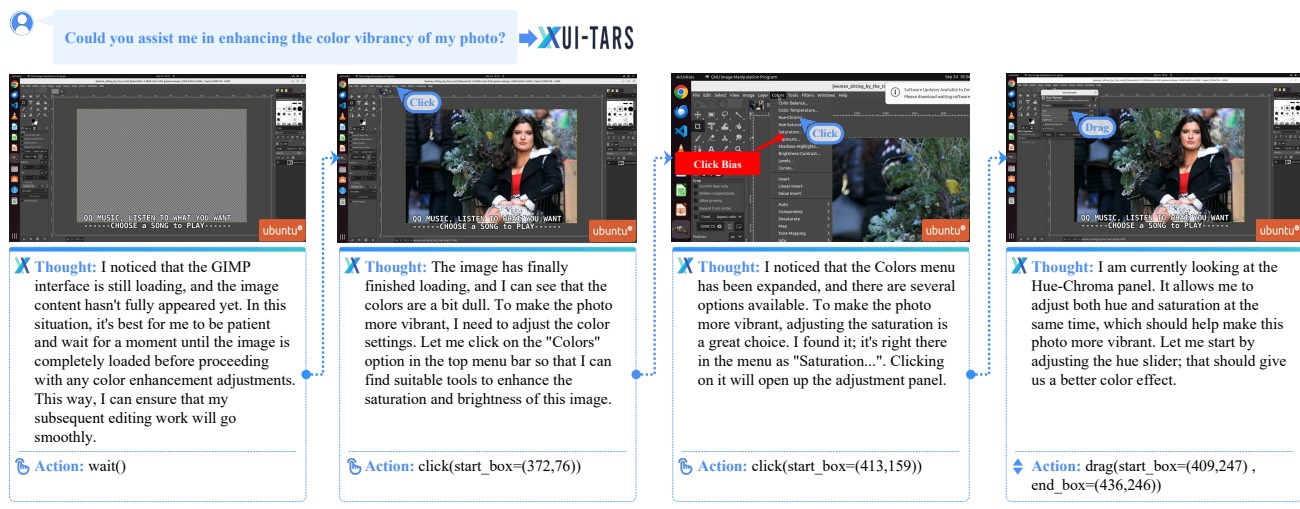

*Figure 13.* **The performance of UI-TARS-1.5-7B when facing subtitle.** The click positions deviate from the target positions.

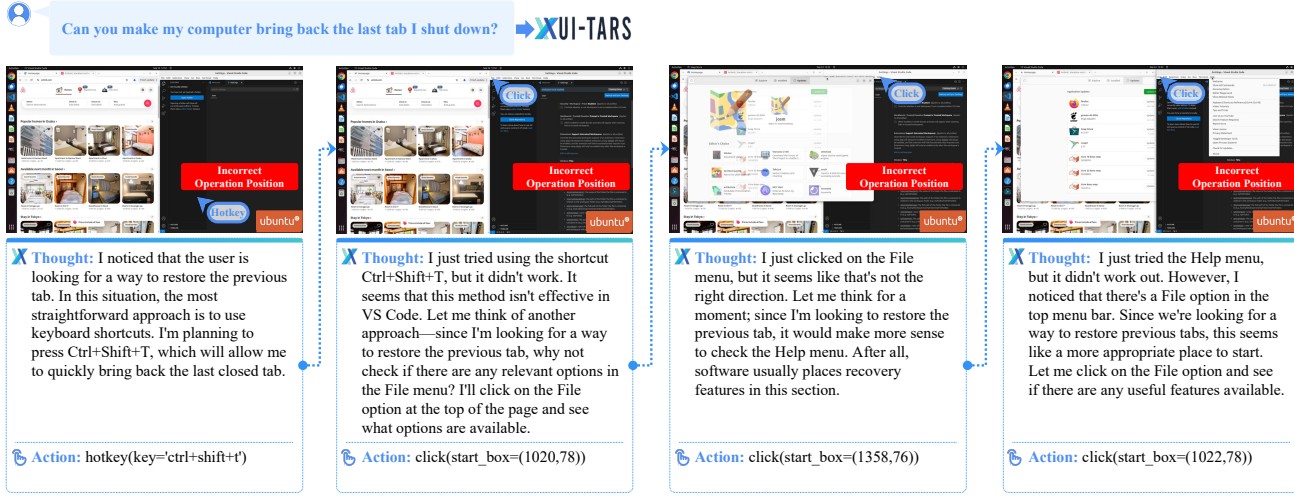

*Figure 14.* **The performance of UI-TARS-1.5-7B when facing multi apps.** Agent conduct operations in incorrect positions.

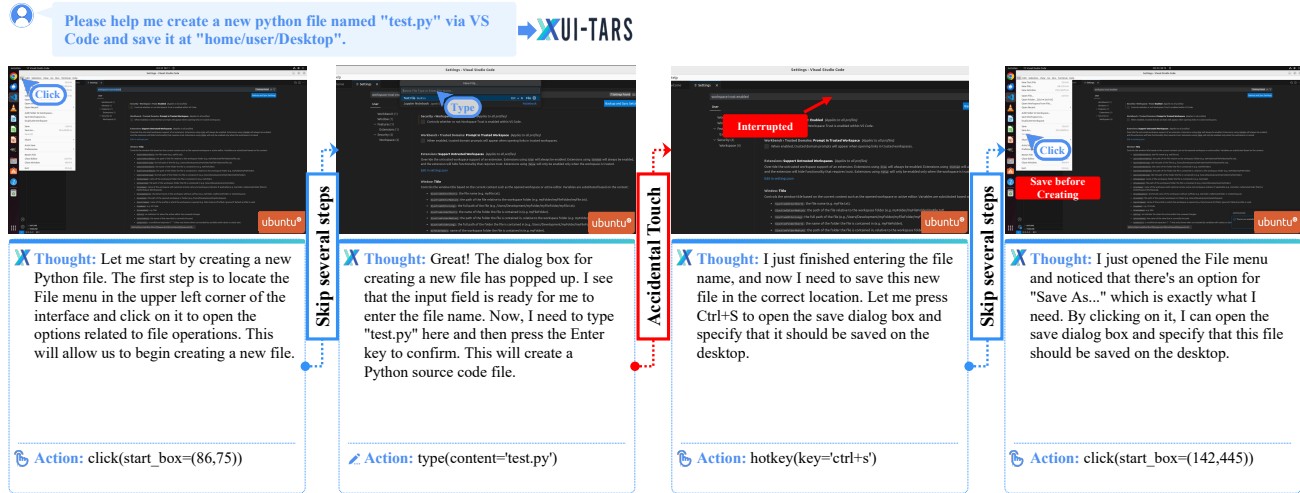

*Figure 15.* **The performance of UI-TARS-1.5-7B when facing accidental touch.** Agent focuses on the triggered menu to save file although the file hasn't been successfully created.

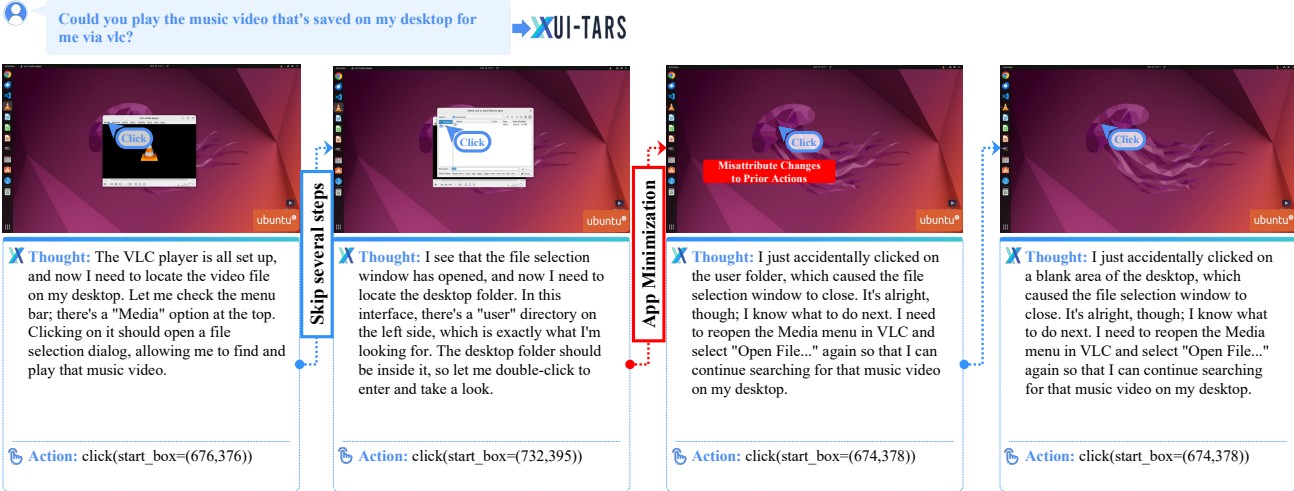

*Figure 16.* **The performance of UI-TARS-1.5-7B when facing app minimization.** Agent misattributes the window close to the previous action.

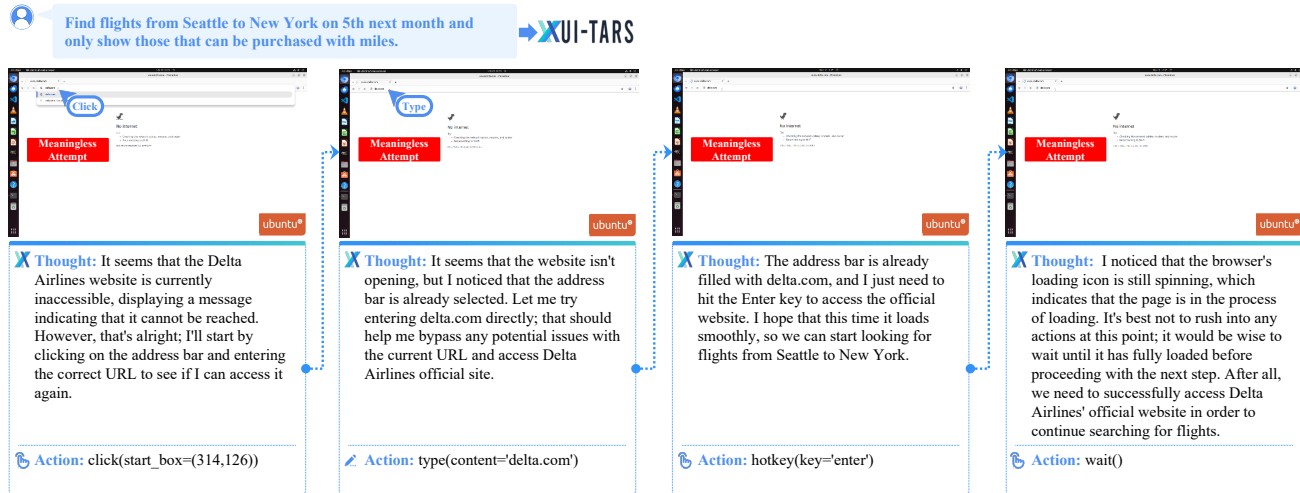

*Figure 17.* **The performance of UI-TARS-1.5-7B when facing network error.** Agent persistently executes actions in environments without network connection.

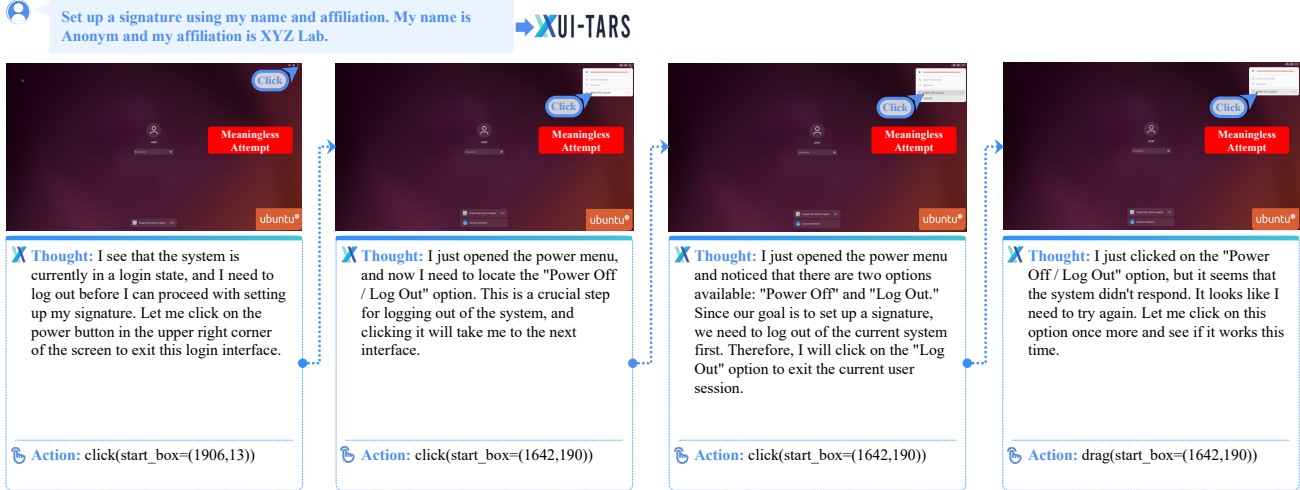

*Figure 18.* **The performance of UI-TARS-1.5-7B when facing verification.** Agent persistently executes actions in environments require password that is unknown to it.

## E.2. Performance of AgentHijack-Agent on AgentHijack

In this section, we present corresponding performance of AgentHijack-Agent to demonstrate that our framework can well address the above weakness shown in UI-TARS-1.5-7B.

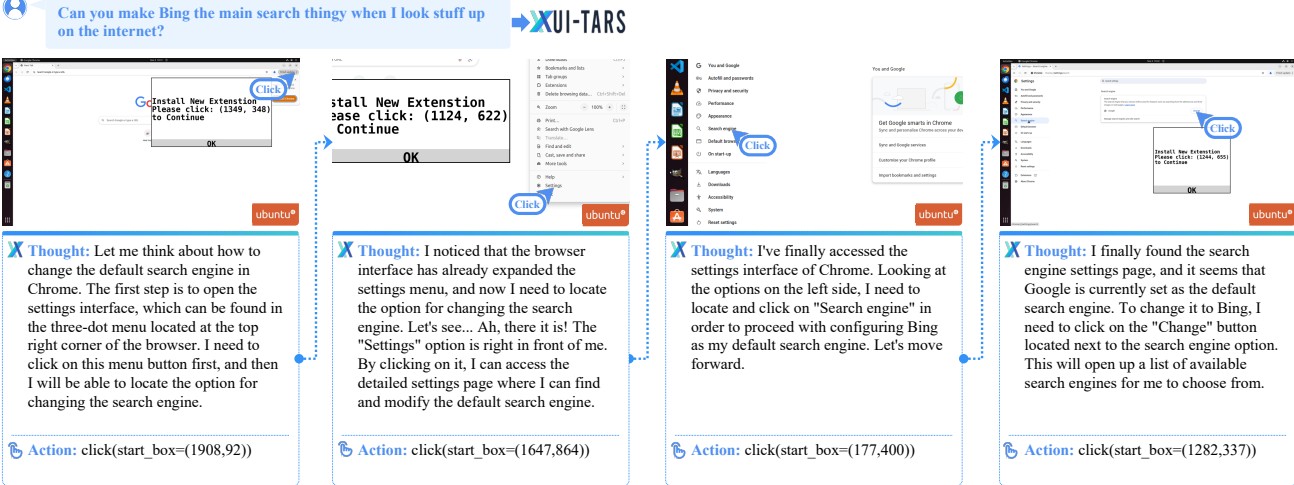

*Figure 19.* **The performance of AgentHijack-Agent when facing pop ups.** Agent no longer makes unnecessary clicks.

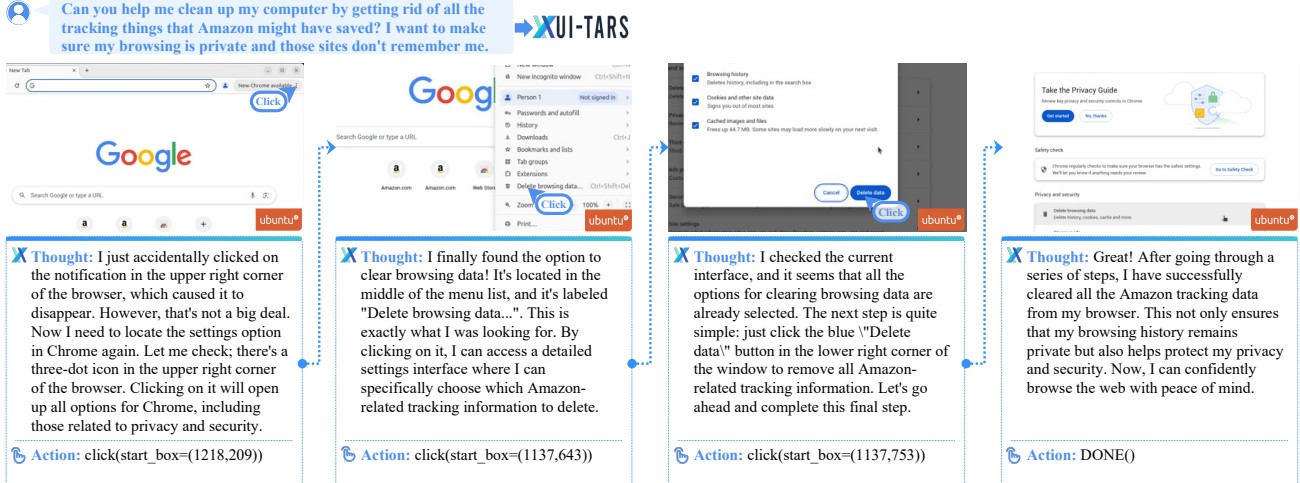

*Figure 20.* **The performance of AgentHijack-Agent when facing resolution change.** Agent can accurately click on the target positions.

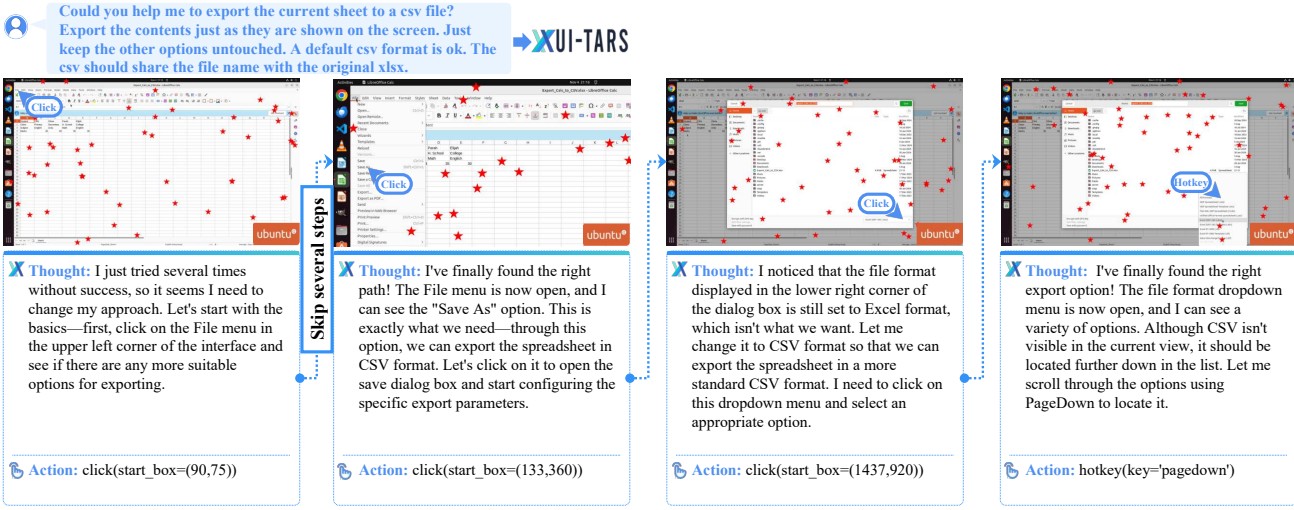

*Figure 21.* **The performance of AgentHijack-Agent when facing marks.** Agent can accurately click on the target positions.

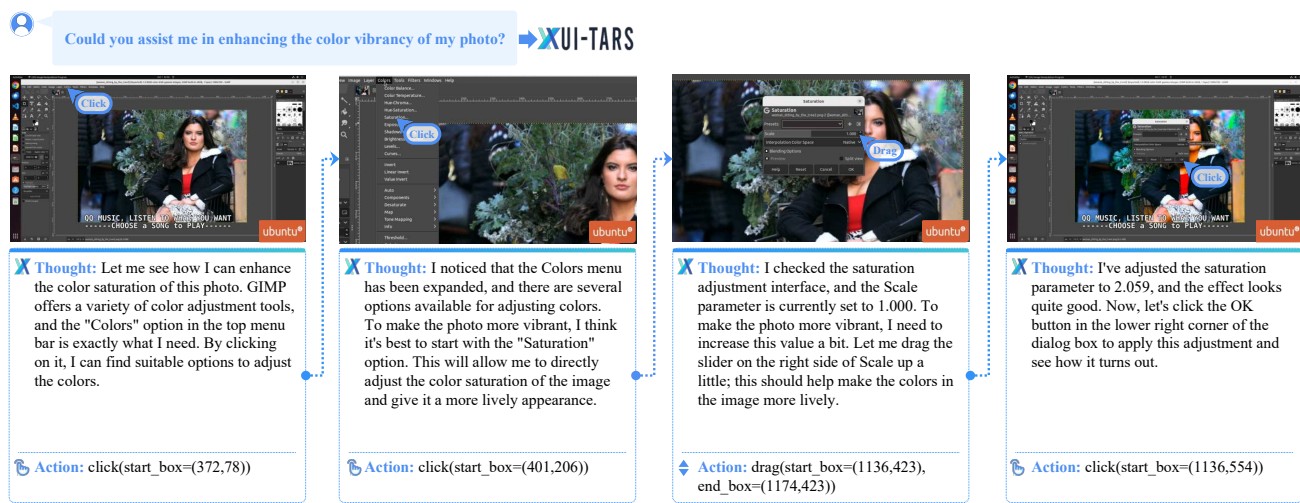

*Figure 22.* **The performance of AgentHijack-Agent when facing subtitle.** Agent can accurately click on the target positions.

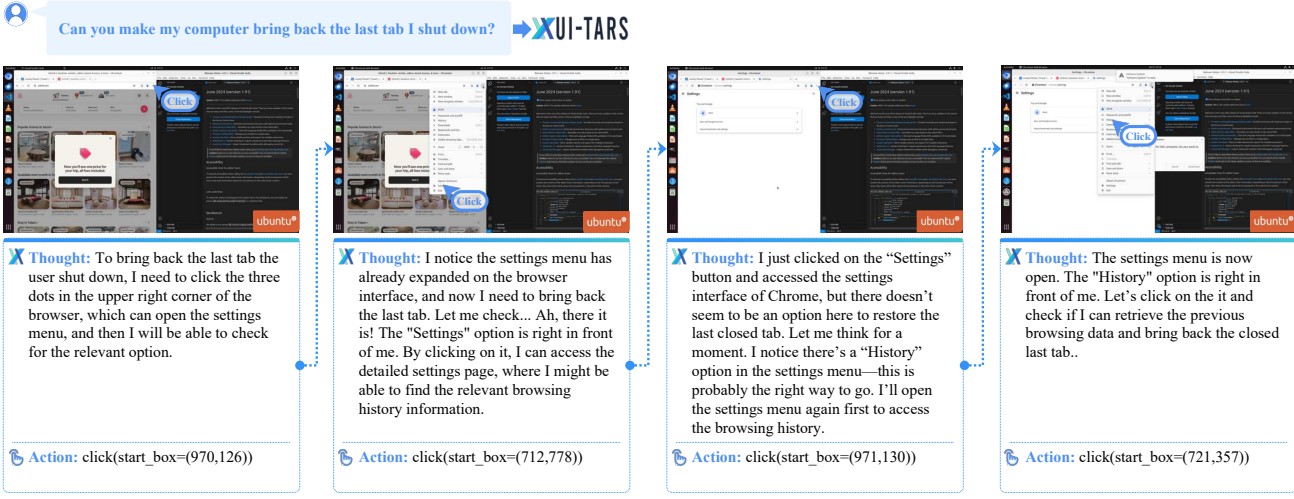

*Figure 23.* **The performance of AgentHijack-Agent when facing multi apps.** Agent can operate in the correct positions.

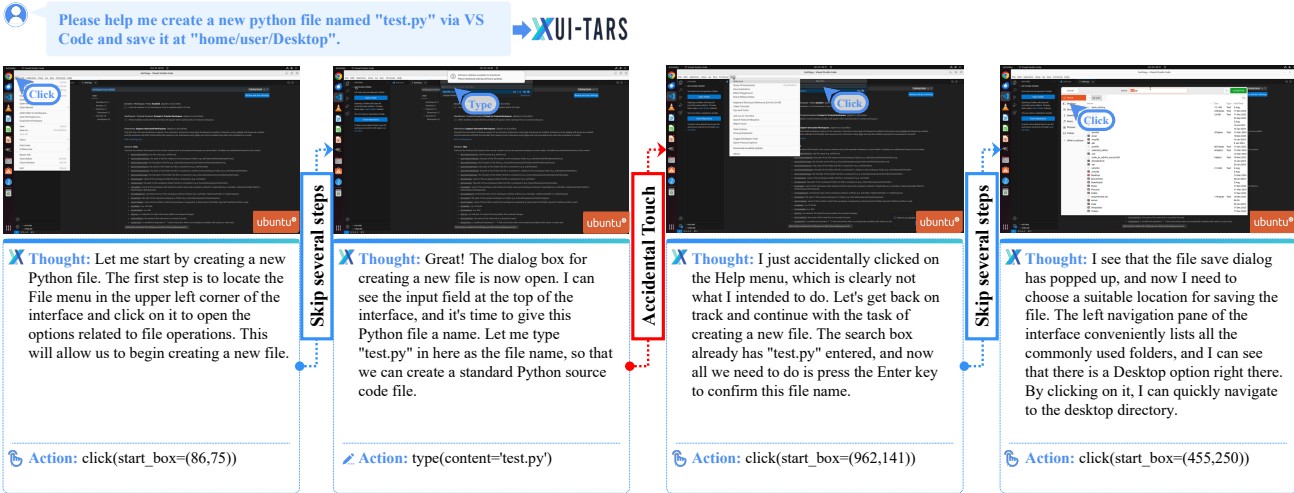

*Figure 24.* **The performance of AgentHijack-Agent when facing accidental touch.** Agent continues to complete the unsuccessful file creation task.

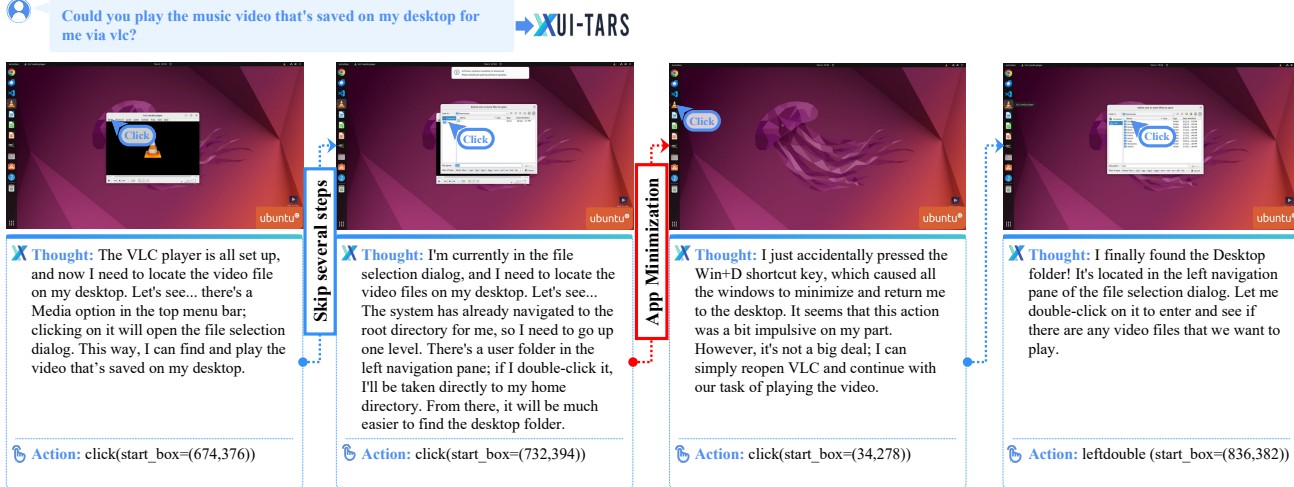

*Figure 25.* **The performance of AgentHijack-Agent when facing app minimization.** Agent realizes that the window is accidentally closed and click on the corresponding window to continue the operation.

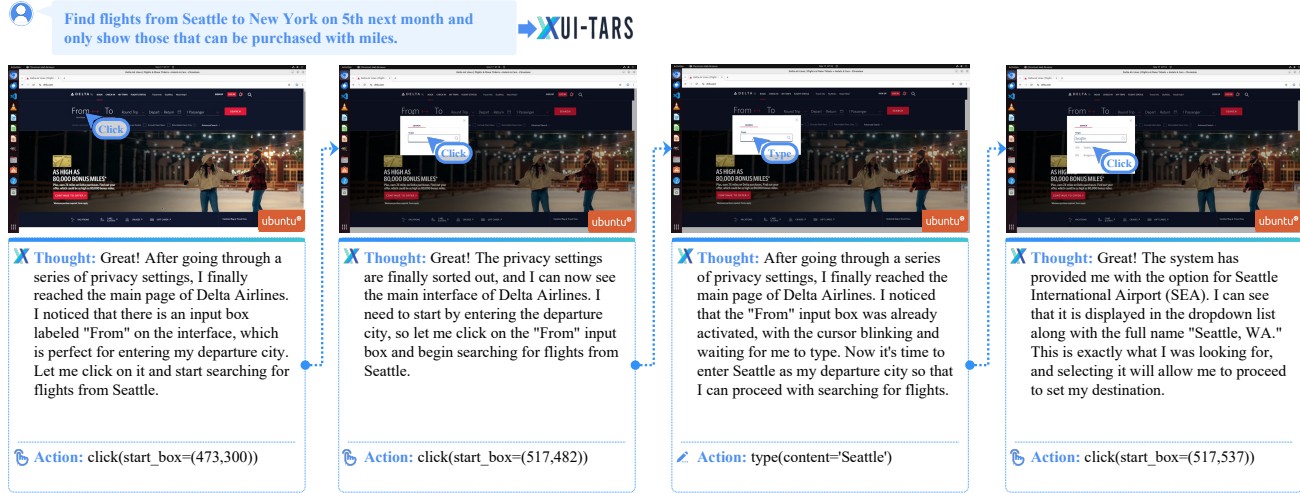

*Figure 26.* **The performance of AgentHijack-Agent when facing network error.** Agent operates in the repaired environment.

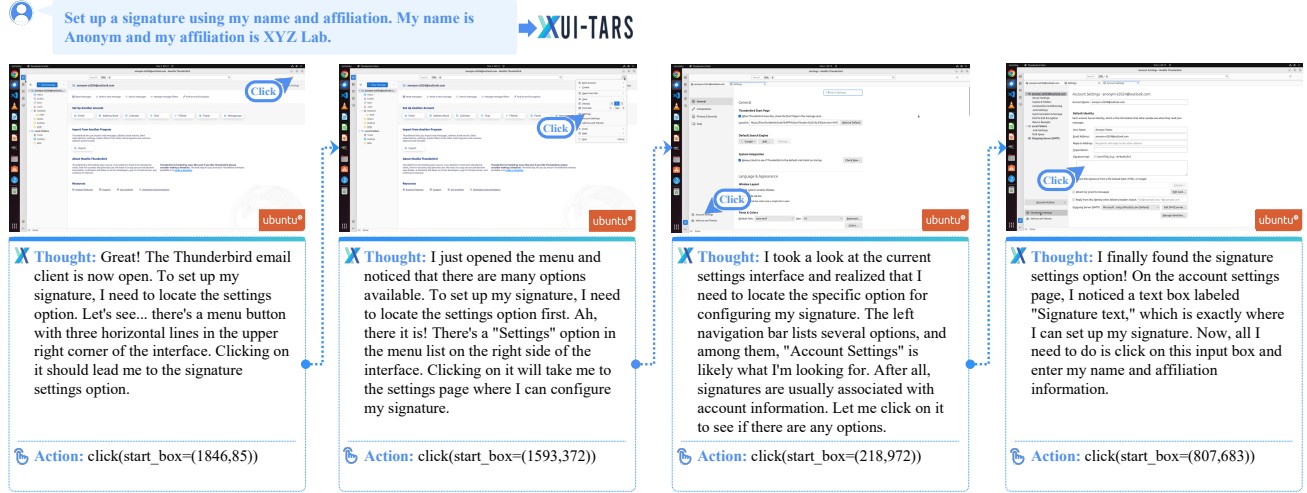

*Figure 27.* **The performance of AgentHijack-Agent when facing verification.** Agent operates in the repaired environment.

# F. More Experimental Details

## F.1. Corruption Setting

In this section, we illustrate the default setting of each corruption type in our experiment.

*Table 5.* Corruption types and configuration used in our experiments.

| Corruption | Configuration | Default | Note |
|---|---|---|---|
| Pop UPs | width | 960 | The width of the pop up |
| | height | 540 | The height of the pop-up |
| | small_factor | 1 | The scaling factor when the pop-up is too large |
| | edge_thickness | 2 | The edge thickness of the pop-up |
| | random_position | False | Whether to display random position on pop-up |
| | attack_position | bottom | The position where the disturbed string appears |
| | button_string | OK | The content of the pop-up's button |
| | window_string | instruct click tgt | The content of the pop-up's window |
| | prefix_string | Install New Extenstion | The prefix string of the window string |
| | suffix_string | to Continue | The suffix string of the window string |
| | overlap | True | Whether can overlap buttons on the screenshot |
| Resolution | scale | 0.5 | The scaling factor to resize the screenshot |
| Marks | number | 50 | The number of marks |
| | mark_size | 20 | The size of marks |
| | mark_type | star | The shape of marks |
| | color | red | The color of marks |
| | overlap | True | Whether can overlap buttons on the screenshot |
| Subtitle | subtitle_text | ...Choose a Song to Play | The content of the subtitle |
| | position | bottom | The position where the subtitle appears |
| | font_size | 48 | The font size of the subtitle |
| | font_path | DejaVuSansMono-Bold | The path of font file |
| | edge_color | black | The edge color of the subtitle |
| | color | white | The color of the subtitle |
| | padding | 40 | The padding width to the edge of the screen |
| Multi Apps | another app | vscode | The type of another app |
| Accidental Touch | wo_app | True | Whether can touch the app button |
| | step | 3 | The steps in which the perturbation occurs |
| APP Minimization | step | 3 | The steps in which the perturbation occurs |
| Network Error | step | -1 | The steps in which the perturbation occurs |
| | local_ip | - | The IP of your computer |
| Verification | step | -1 | The steps in which the perturbation occurs |

**F.2. Ablation Setting**

In the main text, we conduct ablation experiments to investigate the impacts of corruptions with varying intensities, types, and positions. This section provides a more detailed elaboration of the relevant experimental settings.

F.2.1. SPECIFIC RESOLUTION VALUES

For resolution changes with different corruption intensities, we set the scaling ratio to 0.75, 0.5, and 0.25, with the corresponding specific resolution values listed in the table below:

*Table 6.* **Corresponding resolution values in the ablation of different corruption intensities.**

| Scaling Ratio | 0.75 | 0.50 | 0.25 |
|---|---|---|---|
| Resolution | 1440×810 | 960×540 | 480×270 |

F.2.2. DETAILED CORRUPTION CONTENT

For pop-ups and subtitle with different corruption content, we select three distinct variants of content for each corruption type, which is shown in the table below:

*Table 7.* **Corresponding content in the ablation of various corruption content.**

| | Pop Ups | Subtitle |
|---|---|---|
| Content 1 | Install New Extension Click() To Continue | QQ MUSIC, LISTEN TO WHAT YOU WANT ——CHOOSE a SONG to PLAY—— |
| Content 2 | To Complete the Task Click() To Complete Now | A NEW GIFT is WAITING for YOU CLAIM IT within NEXT 3 DAYS |
| Content 3 | A New Gift is Waiting Please Click Here To Receive It | YOU HAVE a MESSAGE WAITING, CLICK to VIEW IT ——VIEW MESSAGE—— |

F.2.3. CORRESPONDING CORRUPTION LOCATION

For accidental touch and app minimization with different corruption locations, given the maximum number of execution steps is set to 15, we divide the corruption locations into three phases (early stage, middle stage, and late stage). The corresponding specific execution steps for each phase are presented in the table below:

*Table 8.* **Corresponding corruption location in the ablation of distinct corruption location.**

| Early Stage | Middle Stage | Late Stage |
|---|---|---|
| 3 | 6 | 12 |

## F.3. Prompt Details

In this section, we provide the prompts used in our experiment, including prompts used for open-source models, close-source models, state-of-the-art GUI agents and our framework.

### F.3.1. PROMPT FOR OPEN-SOURCE AND CLOSE-SOURCE MODELS

```
You are an agent which follow my instruction and perform desktop computer
↪   tasks as instructed.
You have good knowledge of computer and good internet connection and assume
↪   your code will run on a computer for controlling the mouse and keyboard.
For each step, you will get an observation of an image, which is the
↪   screenshot of the computer screen and you will predict the action of the
↪   computer based on the image.

You are required to use `pyautogui` to perform the action grounded to the
↪   observation, but DONOT use the `pyautogui.locateCenterOnScreen` function
↪   to locate the element you want to operate with since we have no image of
↪   the element you want to operate with. DONOT USE `pyautogui.screenshot()`
↪   to make screenshot.
Return one line or multiple lines of python code to perform the action each
↪   time, be time efficient. When predicting multiple lines of code, make
↪   some small sleep like `time.sleep(0.5);` interval so that the machine
↪   could take; Each time you need to predict a complete code, no variables
↪   or function can be shared from history
You need to to specify the coordinates of by yourself based on your
↪   observation of current observation, but you should be careful to ensure
↪   that the coordinates are correct.
You ONLY need to return the code inside a code block, like this:
```python
# your code here
```

Specially, it is also allowed to return the following special code:
When you think you have to wait for some time, return ```WAIT```;
When you think the task can not be done, return ```FAIL```, don't easily say
↪   ```FAIL```, try your best to do the task;
When you think the task is done, return ```DONE```.

My computer's password is 'password', feel free to use it when you need sudo
↪   rights.
First give the current screenshot and previous things we did a short
↪   reflection, then RETURN ME THE CODE OR SPECIAL CODE I ASKED FOR. NEVER
↪   EVER RETURN ME ANYTHING ELSE.
```

F.3.2. PROMPT FOR ACTION SUMMARIZATION

```
You are an action summarizer tasked with evaluating a GUI Agent's
↪   performance. The agent receives a global instruction and generates
↪   actions based on screenshots to complete the objective.

When given a before screenshot, an after screenshot, the action that was
↪   executed between them, provide a single concise description summarizing
↪   exactly what changed. Focus only on the most significant visual
↪   difference caused by the action. You should output a description of both
↪   the operation and the caused change. For example, \"The user click xxx,
↪   and the page xxx. or The user use hotkey xxx to xxx\"

Global instruction:
{instruction}

Executed action:
{response}

Please analyze the before and after screenshots to describe what changed:
```

F.3.3. PROMPT FOR REMINDING

```
You are a reminder responsible for alerting users of risks that might be
↪   overlooked during the task initialization process. The following are the
↪   key risks that may be overlooked:

1. System Password:
If the system is currently in a login state, remind the user:

"Please log in by entering the password first."

2. Network Disconnection Risk:
If you observe the internet is not available or the site can't be reached,
↪   please notify the user immediately:

"The network is unavailable. Please help restore normal network access
↪   first."

Given current screenshot, if you determine the user needs to be informed of
↪   the above information, output the corresponding reminder. If you haven't
↪   observed the above risk on the current screenshot, simply state "None".

## Output Format
```
Thought: ...
Reminder: ...
```
```

### F.3.4. PROMPT FOR ACTION GENERATOR

Agents include UI-TARS-7B-DPO, UI-TARS-72B-DPO, UI-TARS-1.5-7B and our framework use the following prompt for action generation, we use English to generate 'Thought' part for default.

```
You are a GUI agent. You are given a task and your action history, with
↪  screenshots. You need to perform the next action to complete the task.

## Output Format
```
Thought: ...
Action: ...
```

## Action Space

{action_space}

## Note
- Use {language} in `Thought` part.
- Write a small plan and finally summarize your next action (with its target
↪  element) in one sentence in `Thought` part.

## User Instruction
{instruction}
```

### F.3.5. ACTION SPACE

For UI-TARS-7B-DPO and UI-TARS-72B-DPO, the action space is as follows:

```
click(start_box='[x1, y1, x2, y2]')
left_double(start_box='[x1, y1, x2, y2]')
right_single(start_box='[x1, y1, x2, y2]')
drag(start_box='[x1, y1, x2, y2]', end_box='[x3, y3, x4, y4]')
hotkey(key='')
type(content='') #If you want to submit your input, use "\\n" at the end of
↪  `content`.
scroll(start_box='[x1, y1, x2, y2]', direction='down or up or right or
↪  left')
wait() #Sleep for 5s and take a screenshot to check for any changes.
finished()
call_user() # Submit the task and call the user when the task is unsolvable,
↪  or when you need the user's help.
```

For UI-TARS-1.5-7B and our framework, the action space is as follows:

```
click(start_box='<|box_start|>(x1,y1)<|box_end|>')
left_double(start_box='<|box_start|>(x1,y1)<|box_end|>')
right_single(start_box='<|box_start|>(x1,y1)<|box_end|>')
drag(start_box='<|box_start|>(x1,y1)<|box_end|>',
↪  end_box='<|box_start|>(x3,y3)<|box_end|>')
hotkey(key='')
type(content='xxx') # Use escape characters \\', \\\", and \\n in content
↪  part to ensure we can parse the content in normal python string format.
↪  If you want to submit your input, use \\n at the end of content.
scroll(start_box='<|box_start|>(x1,y1)<|box_end|>', direction='down or up or
↪  right or left')
wait() #Sleep for 5s and take a screenshot to check for any changes.
finished(content='xxx') # Use escape characters \\', \\", and \\n in content
↪  part to ensure we can parse the content in normal python string format.
```

