# OpenReview forum: "AgentHijack: Benchmarking Computer Use Agent Robustness to Common Environment Corruptions"
_ICML.cc/2026/Conference — ICML 2026 regular_

### Official Review · Reviewer_4hsu · 2026-02-19

**Soundness:** 3
**Presentation:** 3
**Significance:** 3
**Originality:** 3
**Overall Recommendation:** 5
**Confidence:** 3

**Summary:**

The paper introduces AgentHijack, a benchmark designed to evaluate the robustness of MLLM-powered GUI agents against common environmental corruptions. Evaluations of existing agents on this benchmark reveal their fragility to environmental corruptions. To enhance the robustness against such corruptions, this paper further proposes AgentHijack-Agent, a multi-agent framework that integrates an action generator with enhanced grounding capabilities and an onlooker responsible for behavior summarization and environment checking. Experimental results verify the effectiveness of this framework.

**Compliance With Llm Reviewing Policy:**

Affirmed.

**Final Justification:**

The author basically addressed my concerns during rebuttal. Therefore I decide to raise my score to 5 accordingly.

**Key Questions For Authors:**

Overall, I lean positive on this paper. If the rebuttal is convincing, I would raise my score.

1. What is the detailed statistics of this benchmark?
2. Discuss the latency of the onlooker.

**Limitations:**

Yes

**Strengths And Weaknesses:**

## Strengths
1. As GUI agents move toward real-world deployment, robustness against unpredictable environmental corruptions (e.g., pop-ups, resolution shifts) is a critical prerequisite for reliability. This work addresses a vital yet under-explored gap between *clean* laboratory benchmarks and *noisy* real-world applications.
2. This paper proposes AgentHijack benchmark that introducing 9 configurable common corrutions on desktop platforms. Evaluations of existing agents on this benchmark reveal their fragility to environmental corruptions.
3. To enhance the robustness against such corruptions, this paper further proposes AgentHijack-Agent, a multi-agent framework that integrates an action generator with enhanced grounding capabilities and an onlooker responsible for behavior summarization and environment checking.
4. Experimental results on main body and appendix verify the effectiveness of this framework.

## Weaknesses
1. Insufficient benchmark statistics in main text. The current manuscript omits crucial quantitative details of the benchmark in the primary sections. To improve self-containment, Section 3.3 should include a brief summary of the dataset scale, train/test splits, and the distribution of corruption types.
2. Omission of latency analysis. As a multi-agent framework, AgentHijack-Agent inherently introduces latency from the onlooker. The paper lacks a discussion on the trade-off between increased robustness and inference latency. Analyzing the temporal overhead of the onlooker module relative to the total execution time is essential for assessing its real-time utility.
3. Suboptimal figure readability. The presentation of experimental results could be improved. In Figures 4, 5, and 6, the legends overlap with the data points or trend lines. Relocating the legends to the top of the figures or using a shared legend would enhance the visual clarity.

---

> ### Author Rebuttal · Authors · 2026-03-29
>
> Dear Reviewer 4hsu,
>
> Thank you for your valuable comments. In response to your questions, we will address them point by point, and our replies are as follows:
>
> >Q1. Insufficient benchmark statistics in main text. The current manuscript omits crucial quantitative details of the benchmark in the primary sections. To improve self-containment, Section 3.3 should include a brief summary of the dataset scale, train/test splits, and the distribution of corruption types.
>
> Thank you for your valuable comments. We summarize the task distribution based on app domain and operation types as follows:
>
> |App Domain|Professional|Workflow|Daily|OS|Office|
> |-|-|-|-|-|-|
> |Operation Types|Image Operations (26/7.0%)|File Operations (30/8.1%)|Settings (21/5.7%)|Terminal (7/1.9%)|Visualization (7/1.9%)|
> ||Configuration (14/3.8%)|Multimedia (17/4.6%)|Info Query (15/4.1%)|Settings (9/2.4%)|Processing (26/7.0%)|
> ||Code Assist (9/2.4%)|Data Analysis (33/8.9%)|Shopping (10/2.7%)|Files (8/2.2%)|Table Formatting (14/3.8%)|
> |||Miscellaneous (21/5.7%)|Account Operations (6/1.6%)||Slide Settings (15/4.1%)|
> ||||Email Operations (9/2.4%)||Slide Editing (32/8.7%)|
> ||||Vedio Control (17/4.6%)||Document Settings (6/1.6%)|
> ||||||Document Editing (17/4.6%)|
> |Total Number|49/13.3%|101/27.4%|78/21.1%|24/6.5%|117/31.7%|
>
> **Professional tasks** involve operations on specialized software such as VSCode and GIMP; **Workflow tasks** consist of interactions across different applications; **Daily tasks** cover operations on commonly used software like Chrome; **OS tasks** focus on manipulations of the computer operating system; **Office tasks** include handling office suites such as Excel and PowerPoint. Different types of corruptions will be uniformly applied across all tasks, ensuring that **AgentHijack can evaluate the corruption robustness of GUI agents on a wide range of computer-based tasks.**
>
> >Q2. Omission of latency analysis. As a multi-agent framework, AgentHijack-Agent inherently introduces latency from the onlooker. The paper lacks a discussion on the trade-off between increased robustness and inference latency. Analyzing the temporal overhead of the onlooker module relative to the total execution time is essential for assessing its real-time utility.
>
> Thanks for your advice. We measure the latency of the onlooker as below.
> ||device|latency
> |-|-|-
> |summary|1*A100|1.73s
> |environment check|1*A100|1.42s
>
> Compared with the average total task latency (321.19s on an A100 GPU), the onlooker only **incurs an additional overhead of about 5%** (averaging 10 steps, with environment check executed only before the first step), while achieving an average performance improvement of 20%, which is acceptable.
>
> The onlooker module processes only a small number of tokens at each step, while the action generator accumulates an increasingly long history trajectory as execution proceeds, resulting in much higher inference latency. Therefore, compared with the action generator, the onlooker module does not introduce excessive or unacceptable overhead.
>
> >Q3. Suboptimal figure readability. The presentation of experimental results could be improved. In Figures 4, 5, and 6, the legends overlap with the data points or trend lines. Relocating the legends to the top of the figures or using a shared legend would enhance the visual clarity.
>
> Thank you for your helpful suggestion. Our original intention was to maintain a consistent layout across all figures, so we placed all legends uniformly in the bottom-right corner. We will follow your advice and adjust the legend positions to avoid occlusion.
>
> Finally, we sincerely thank you again for your professional comments and the precious time you have devoted. If you have any other questions or need further discussion, **you are welcome to raise them at any time**, and we will spare no effort to provide you with satisfactory answers.

---

> > ### Author Rebuttal · Reviewer_4hsu · 2026-04-01
> >
> > My major concens are basically addressed in this rebuttal. I will adjust the score accordingly.

---

> > > ### Author Response · Authors · 2026-04-01
> > >
> > > Dear Reviewer 4hsu,
> > >
> > > **We’re glad to hear that your concerns have been addressed, and your recognition of our work means a lot to us.** If you have any other questions, please don’t hesitate to let us know, we will try our best to address your concerns.
> > >
> > > **Thank you again for your valuable time and comments.**

---

### Official Review · Reviewer_NM8M · 2026-03-11

**Soundness:** 3
**Presentation:** 3
**Significance:** 3
**Originality:** 3
**Overall Recommendation:** 4
**Confidence:** 4

**Summary:**

This paper introduces AgentHijack to examine the robustness of multimodal LLMs against common corruptions, including three categories covering perturbations on environment states, observation function and state transition process. To solve the performance degradation caused by these corruptions, they propose AgentHijack-Agent, which integrates an onlooker to check initial states and summarize environment changes and an action generator as the main actor. The experiments show an overall improvement of the proposed framework on the benchmark performance.

**Compliance With Llm Reviewing Policy:**

Affirmed.

**Final Justification:**

I thank authors for the further clarification, my concerns were partially solved and I will maintain the score

**Key Questions For Authors:**

See Weaknesses for most questions.

1. Even under a clean environment, the task success rate is low. The corruption data augmentation seems to also drag up this clean performance. What is the bottleneck of baseline utility? And how might future iterations of GUI agents address this?

**Limitations:**

The limitations are not adequately discussed in the paper. The authors should add a section to acknowledge the limitations of the proposed benchmark and framework, e.g. the benchmark scope, the computational and latency overhead introduced by the onlooker agent, potential adversarial exploitation etc.

**Strengths And Weaknesses:**

Strengths:
- The robustness of computer use agents against common corruptions is a realistic and important problem.
- The paper formalizes the corruption scope in a clean and complete way.
- The benchmark covers a broad range of configurable corruptions to reflect real-world environments.
- The proposed framework improves the overall performance of agents against most corruptions.

Weaknesses:
- While common corruptions are important and more likely to happen in reality, the adversarial corruptions can also be fit into the perturbation scope modeled in the paper (e.g. an adversarial pop-up and a benign pop-up manipulate the POMDP in the exact same way), the paper didn’t justify the reason of excluding adversarial corruptions.
- A terminology confusion: in 3.1, the task is defined as $x \in D$ where $D$ is the task set and $y$ is the task objective (given $s_T = y$, $y$ should be $\in S$), but later in 3.2, the subscript shows $(x,y) \in D$.
- My main concern on the proposed framework: the baseline model shows a 5.47% performance drop. The proposed framework, though improving the success rate of the basic model by 4.15%, still presents a 4.91% performance drop compared to the performance on clean data. It seems only to show the effectiveness of data augmentation on improving the overall performance, but the 0.56% reduction in the robustness gap is hard to showcase that the robustness has improved significantly.
    - Furthermore, looking at Table 2, the most degraded aspects are: pop-ups, resolution issues, multi-app handling, and verification. Only pop-ups and verification are well-solved, as the onlooker handles the verification and omits pop-ups during summarization, but resolution and multi-app issues still cause large performance drop. Figure 6 also supports this claim (onlooker mainly solves verification).
    - Additionally, for the ablation studies of the onlooker, did you completely remove the onlooker or have you tried to add onlooker's role or prompts (like checking initial environment and highlight environment changes) to the action generator? I'm curious about whether the role separation is the key or a more fine-grained task description also helps.
- Other minor issues:
    - Figure 2 presents many examples but it is hard to see the categorization at a glance. It would be great to have group structure, e.g. each row represents a category of corruptions.
    - Figure 3 lacks a clear highlight of how the new framework changes the specific behavior of the agents compared to the baseline failure modes.

---

> ### Author Rebuttal · Authors · 2026-03-29
>
> Dear Reviewer NM8M,
>
> Thank you for the valuable comments. We will address the questions point by point:
>
> >Q1. didn’t justify the reason of excluding adversarial corruptions
>
> Thank you for your comment. In fact, **we explained our rationale for excluding adversarial robustness in Sec. 2.2 and 3.2 of the main paper.** Prior works such as RiOSWorld have already explored evaluating the adversarial robustness of GUI agents, yet the assessment of common corruptions remains largely unexplored. To fill this gap, we propose AgentHijack. That's why we exclude adversarial corruptions
>
> >Q2. A terminology confusion...
>
> Thank you for pointing out this issue. We will unify the expressions in the revised version
>
> >Q3. The 0.56% reduction in the robustness gap is hard to showcase that the robustness has improved significantly. Furthermore, resolution and multi-app issues still cause large performance drop. Additionally, for the ablation studies of the onlooker, did you completely remove the onlooker or ...
>
> We thank the reviewer for the thorough analysis. We address each point below.
> 1. For the robustness improvement. As can be observed, our framework achieves **consistently superior performance on both clean and corrupted data**, indicating that our approach yields more generalizable performance gains. We respectfully argue that evaluating the effectiveness of robustness improvements solely based on the robustness gap may lead the model to overfit to corrupted data, thereby harming its general-purpose performance.
> 2. For certain challenging corruptions. We would like to provide further analysis: Resolution changes fundamentally alter the spatial mapping between predicted coordinates and actual on-screen positions, which is a core issue at the perceptual level and requires the agent to recalibrate its coordinate system. Given that current models are mostly pre-trained only on fixed-resolution screenshots, we believe this problem calls for improvements in foundational model pre-training, which represents an important future direction enabled by our benchmark. Multi-application interference requires correctly identifying and manipulating task-relevant windows among multiple overlapping applications. Although our framework yields certain improvements, this problem demands higher-level scene understanding beyond what action summaries can provide, posing stronger requirements for the model’s reasoning ability. The varying degrees of improvement achieved by our method precisely reveal the deficiencies of current GUI agents and highlight critical directions for future reinforcement. We argue that this constitutes a novel insight contributed by AgentHijack. We will incorporate this discussion into the revised manuscript
>
> 3. For the onlooker ablation. Yes, we **completely removed the onlooker component and did not further elaborate on the task descriptions.** As we think that ensuring consistent task descriptions is essential for a fair comparison
>
> >Q4. Some readability issues in Figures 2 and 3
>
> Thanks for your advice, we will refine the figures to improve readability in the refined version
>
> >Q5. What is the bottleneck of baseline utility? And how might future iterations of GUI agents address this?
>
> Thank you for your question. We identify the main bottlenecks of the baseline as follows:(a) **Insufficient grounding ability:** Even on clean data, the model can only guarantee accuracy in most cases rather than consistently, and corrupted data further exacerbates this issue. This explains why our method also improves performance on clean data. We believe this requires future improvements in both dataset construction and more advanced RL algorithms.(b) **Limited memory of trajectory history:** This motivates our introduction of the summary mechanism. In the future, we can draw on multi-modal storage and management techniques from the agent memory literature to further help GUI agents retain historical behaviors.(c) **Lack of awareness of initial environmental errors:** This requires an additional dimension of capability for GUI agents. Existing research mainly focuses on improving task performance, while abilities such as environment checking are largely overlooked. Similar to localization, this issue may be addressed by constructing more advanced datasets, enabling foundational pre-trained models to gain capabilities across different dimensions
>
> >Q6. limitation section is needed to discuss...
>
> Thank you for your advice, we will add a section to discuss the limitations of this paper. For the latency introduced by the onlooker, we supplement experiments as follows:
>
> ||device|latency
> |-|-|-
> |summary|1*A100|1.73s
> |environment check|1*A100|1.42s
>
> Compared with the average total task latency (321.19s on an A100 GPU), the onlooker only **incurs an additional overhead of about 5%** (averaging 10 steps, with environment check executed only before the first step), while achieving an average performance improvement of 20%, which is acceptable

---

> > ### Author Rebuttal · Reviewer_NM8M · 2026-04-03
> >
> > I thank the authors for their detailed response. I acknowledge that the robustness improvement on specific categories of corruption are significant, which can be attributed to the new role introduced by the framework, but the overall improvement remains unconvincing since if we just compare two baseline models UI-TARS-7B-DPO and UI-TARS-1.5-7B we can see a similar level of improvement on both clean and most corrupted data, even though the latter was not specifically trained for robustness. And the lack of an ablation study regarding the onlooker's role in a single model makes it difficult to justify the necessity of the current design. That said, I remain generally positive about the paper’s contribution and will maintain my score.

---

> > > ### Author Response · Authors · 2026-04-04
> > >
> > > Dear Reviewer NM8M,
> > >
> > > We are glad that some of your concerns have been addressed. For the remaining questions, we would like to provide further clarification.
> > >
> > > First, regarding the overall performance improvement, we respectfully disagree that the performance gain brought by AgentHijack-Agent is on the same level as that of UI-TARS-1.5-7B. We explain our view from the following aspects:
> > >
> > > - (a) **The degree of improvement varies across different corruption types.** In contrast to our consistent improvements across all corruption categories, UI-TARS-1.5-7B even performs substantially worse than UI-TARS-7B-DPO on certain corruptions—for example, it shows a performance drop of 2.81% on pop-ups.
> > >
> > > - (b) **The nature of the improvement is fundamentally different.** A comparison between Figures 9–17 (failure cases of UI-TARS-1.5-7B) and Figures 18–26 (success cases of our agent) in the appendix demonstrates that the improvement from AgentHijack is not merely a quantitative performance gain, but **a fundamental shift in agent behavior**. Although UI-TARS-1.5-7B achieves notable improvements over UI-TARS-7B-DPO, it still suffers from inaccurate grounding, fragile decision-making under disturbance, and ignorance of environmental errors. In contrast, our method corrects these issues, endowing the model with new capabilities not acquired during pre-training.
> > >
> > > - (c) **The required effort is highly imbalanced.** UI-TARS-1.5-7B relies on significantly more, higher-quality, and larger-scale pre-training data to achieve performance gains comparable to our lightweight approach. This in turn validates that **our method can achieve competitive robustness at a much lower cost.**
> > >
> > > - (d) **The improvements from the two methods are independent and complementary.** Our approach is built upon UI-TARS-1.5-7B as the base model; we do not aim to compete with general capability improvements, but to further enhance robustness on top of them. The ablation experiments in Figure 6 directly show that our method effectively improves the multi-dimensional robustness of the base model, indicating that **our framework can be readily combined with stronger pre-trained models to enable more reliable agent deployment.**
> > >
> > > Second, regarding the ablation study on the onlooker’s role in a single model. We apologize that our response in the previous round may have caused some misunderstanding. We make further illustration here:
> > >
> > > - (a) In our approach, **both the onlooker and the action generator use the same model**, UI‑TARS‑1.5‑7B. When performing different roles, their system prompts are set separately for the onlooker and the action generator. This indicates that **the ablation study shown in Figure 6 has already investigated the onlooker’s role in a single-model setup.** As can be observed, removing the onlooker prompt leads to performance drops on corresponding corruption types, which supports that assigning distinct roles to the same model is reasonable and effective.
> > >
> > > - (b) Furthermore, we also explored **setting the onlooker’s role in different models** in Figure 7. It can be seen that using stronger models as onlooker yield larger performance improvements. However, considering the trade‑off between performance gain and computational overhead, we finally chose to use a single model to serve as both the onlooker and the action generator.
> > >
> > > Finally, thank you once again for your valuable comments and time and please don't hesitate to raise any questions for further discussion. We will spare no effort to address any unclear parts if it needs further clarification.

---

### Official Review · Reviewer_X8CQ · 2026-03-12

**Soundness:** 3
**Presentation:** 3
**Significance:** 3
**Originality:** 3
**Overall Recommendation:** 4
**Confidence:** 4

**Summary:**

This paper looks at how multimodal LLM agents for computer use behave when the environment gets noisy. The kind of noise they care about is realistic things: pop-ups appearing, resolution changing, network problems, accidentally clicking something. All things that happen constantly during normal computer use but that benchmarks tend to ignore.
They build a benchmark called AgentHijack on top of OSWorld. It defines nine corruption types (pop-ups, resolution changes, UI marks, subtitles, multiple windows, accidental clicks, app minimisation, network errors, verification dialogs) and uses them to generate 3,321 corrupted tasks.
They also propose a method, AgentHijack-Agent, with two parts. The first is an action generator trained with Data-Augmented GRPO across corrupted environments to improve grounding. The second is a monitoring module that watches for environment changes, builds a summary of what has happened so far, and validates the environment state before the agent takes an action. The idea is to catch situations where something external has gone wrong before the agent blindly continues.
They test a few open-source and proprietary agents. Performance drops noticeably under corruption. UI-TARS-1.5-7B goes from 24.21% on clean tasks down to 18.74%. Their method brings back about 4.15% average success rate across corruption types. It is not a huge recovery but the problem itself is underexplored so it is a reasonable contribution.

**Compliance With Llm Reviewing Policy:**

Affirmed.

**Key Questions For Authors:**

1. How representative are the proposed corruption types of disturbances encountered in real-world deployments of computer-use agents?
2. To what extent does the observed improvement come from the RL training versus the onlooker module?
3. Since the 128 RL training tasks are sampled from the AgentHijack benchmark itself, how do the authors ensure that the improvements are a genuine reflection of robustness rather than adaptation to the specific task–corruption combinations present in the benchmark?

**Limitations:**

Yes.

The authors discuss limitations related to agent fragility in real-world environments. However, several additional concerns are worth raising.

First, the 128 training tasks used for DA-GRPO are sampled from the AgentHijack benchmark itself, which is also used for evaluation. Although the corruption configurations can vary, the underlying task distribution is shared between training and testing. This makes it difficult to disentangle genuine robustness gains from overfitting to the specific task-corruption combinations encountered during training. A held-out task set or evaluation on a separate environment would have been more convincing.

Second, all nine corruption types are evaluated in isolation. In practice, corruptions co-occur: a pop-up may appear while the resolution is wrong and a network error is active. The paper does not evaluate any combined corruption scenarios, so it remains unclear whether the proposed framework degrades gracefully under simultaneous disruptions or whether the improvements observed in single-corruption settings would carry over.

Third, the onlooker module introduces an additional model forward pass at every step for behaviour summarisation, plus an initial environment check before execution. The paper does not report the computational overhead this entails in terms of wall-clock time or token cost per task. For a framework motivated by real-world deployment, this is a notable omission.

**Strengths And Weaknesses:**

## Strengths
The paper fills a real gap. Most benchmarks for computer-use agents assume clean environments, which is not how people typically use computers. The corruption types they pick are sensible and grounded in things that genuinely happen during normal use. The benchmark itself is configurable, so you can mix and match corruption types and regenerate scenarios, which is good for reproducibility. The failure mode analysis is also useful. They clearly identify where agents break: grounding errors, misattributing environment changes, and failing to notice when the environment was already broken at initialisation. The evaluation covers a decent spread of both open-source and proprietary agents.

## Weaknesses
The method is not very convincing. RL fine-tuning with data augmentation, trajectory summarisation, monitoring are not new, and putting them together does not automatically make a contribution. The paper needed to argue more carefully why this specific combination matters, and it does not really do that. A 4% average improvement in success rate does not help the case either. The corruptions themselves are also limited to UI-level. Latency, dynamic interfaces, or interruptions during longer multi-step tasks are not considered, but those happen just as often. And because the RL training only covers a small subset of tasks, generalisation is unclear (see Limitations). They also only report binary success/failure. If they had looked at partial progress or recovery behaviour they probably would have had a better picture of what the agent is actually doing when things go wrong.

---

> ### Author Rebuttal · Authors · 2026-03-29
>
> Dear Reviewer X8CQ,
>
> Thank you for your valuable comments. In response to your questions, we will address them point by point, and our replies are as follows:
>
> >Q1. How representative are the proposed corruption types of disturbances encountered in real-world deployments of computer-use agents?
>
> Thank you for your question, **all corruption types we adopt are common in daily usage scenarios.** For instance, pop-ups simulate chat or advertisement windows, marks simulate screensaver effects, and subtitles simulate the visual interference introduced by music or video applications, which is also presented in Section 3.3.
>
> >Q2. To what extent does the observed improvement come from the RL training versus the onlooker module?
>
> As shown in the fourth subplot of Figure 6, for marks corruption, RL training yields an improvement of 2.51%, which is higher than the 1.46% gain brought by the onlooker. In contrast, on accidental touch and verification, the onlooker achieves larger gains compared to RL training, with improvements of 1.06% and 8.62% respectively. This aligns with our earlier analysis: RL training is designed to address grounding inaccuracies caused by visual disruptors, while the onlooker specifically targets decision-making interference from unexpected actions and failure to perceive environmental errors induced by the latter two corruption types. This further validates the rationality of our method design.
>
> >Q3. Since the 128 RL training tasks are sampled from the benchmark itself, how to ensure that the improvements are a reflection of robustness rather than adaptation to the specific task–corruption?
>
> Thank you for your valuable comment. We address this concern from three perspectives.
>
> Firstly, in addition to performance improvements under various corruptions, **the RL-finetuned agent also achieves notable gains on clean data**, which is not included in the RL training set. This indicates that the improvement is not domain-specific adaptation, but a general performance enhancement.
>
> Secondly, **we further report performance improvements on tasks outside the training set**, as shown below. It can be observed that the model also obtains gains on unseen tasks for corruption types targeted by RL, such as pop-ups and marks.
>
> |Corruption|Pop ups|Resolution|Marks|Subtitle|Multi Apps|
> |-|-|-|-|-|-|
> |Improvement|+3.57%|+0.42%|+2.16%|+2.29%|+1.76%
>
> Thirdly, **we conduct additional experiments on unseen corruption types and test with mixtures of different corruptions.** The results will be presented together with our response to the next question, where the model still achieves considerable improvements.
>
> Based on these three lines of evidence, we conclude that RL training yields generalizable robustness improvements rather than overfitting to specific corruptions or tasks.
>
> >Q4. all nine corruption types are evaluated in isolation. In practice, corruptions co-occur, it remains unclear whether the proposed framework degrades gracefully under simultaneous disruptions or whether the improvements observed in single-corruption settings would carry over.
>
> To comprehensively evaluate the performance of our method under scenarios with multiple types of corruptions simultaneously, we conducted additional experiments with mixed corruptions. Specifically, we combined three major categories of corruption, i.e., visual disruptors, unexpected operations, and environment errors, for testing. The results are presented below. It can be seen that our framework remains effective even in the presence of multiple concurrent disturbances, which further confirms that the proposed method delivers general and reliable robustness improvements.
>
> |corruption|pop ups+accidental touch|pop ups+network error|accidental touch+network error|
> |-|-|-|-|
> Mode|visual disruptors+unexpected operations|visual disruptors+environment errors|unexpected operations+environment errors|
> UI-TARS-1.5-7B|8.12%|7.78%|16.43%
> Ours|17.09%|15.64%|18.49%
>
> >Q5. the onlooker module introduces an additional model forward pass at every step for behaviour summarisation, plus an initial environment check before execution. The paper does not report the computational overhead. For a framework motivated by real-world deployment, this is a notable omission.
>
> Thanks for your advice. We measure latency of the onlooker as below.
> ||device|latency
> |-|-|-
> |summary|1*A100|1.73s
> |environment check|1*A100|1.42s
>
> Compared with the average total task latency (321.19s on an A100 GPU), the onlooker only **incurs an additional overhead of about 5%** (averaging 10 steps, with environment check executed only before the first step), while achieving an average performance improvement of 20%, which is acceptable.
>
> Finally, we sincerely thank you again for your professional comments and the precious time you have devoted. If you have any other questions or need further discussion, you are welcome to raise them at any time, and we will spare no effort to provide you with satisfactory answers.

---

> > ### Author Rebuttal · Reviewer_X8CQ · 2026-04-03
> >
> > Rebuttal addressed my questions

---

> > > ### Author Response · Authors · 2026-04-03
> > >
> > > Dear Reviewer X8CQ,
> > >
> > > **We are glad to hear that all your concerns have been fully addressed, and we appreciate your recognition of our work.** We would be grateful if you could kindly consider adjusting your score. Thank you once again for your valuable comments and time.
> > >
> > > If you have any additional questions before the end of the discussion period, please feel free to let us know, and **we will try our best to resolve your concerns.**

---

### Official Review · Reviewer_G5fH · 2026-03-14

**Soundness:** 2
**Presentation:** 3
**Significance:** 2
**Originality:** 2
**Overall Recommendation:** 2
**Confidence:** 4

**Summary:**

This paper introduces AgentHijack, a benchmark to test how well computer-use agents (powered by MLLMs) handle common real-world disruptions that aren't malicious attacks. They define 9 types of corruptions: pop-ups, resolution changes, screen marks, subtitles, multiple apps running, accidental touches, app minimization, network errors, and login verification screens. They apply these corruptions to OSWorld tasks, creating 3,321 test cases. The paper also proposes AgentHijack-Agent, which combines an action generator trained with Data-Augmented GRPO and an "onlooker" module that summarizes actions and checks for environment errors.

**Compliance With Llm Reviewing Policy:**

Affirmed.

**Key Questions For Authors:**

see Strengths And Weaknesses

**Limitations:**

see Strengths And Weaknesses

**Strengths And Weaknesses:**

Strengths:
- practical and real world setting;
- Systematic corruption taxonomy (9 corruption types are well-categorized)
- Thorough experiments and nice case study on fail cases


Weaknesses:
- My biggest concern is significance. The benchmark contribution is somewhat engineering-focused - they're applying image corruptions and simulated disruptions to an existing benchmark (OSWorld). The corruption types, while practical, are straightforward to implement (resize images, overlay pop-ups, disconnect network). It's not clear what new scientific insights we gain beyond "agents fail when things are messy" - which is pretty expected.
- AgentHijack-Agent combines existing pieces: DA-GRPO is just GRPO with data augmentation from corrupted environments; The onlooker is essentially calling another LLM to summarize what happened; Experience replay buffer is borrowed from ARPO. None of these components are new. The contribution feels incremental.
- Some corruptions feel unrealistic:
  1. Random star/rectangle marks appearing on screen (when does this happen in real life?)
  2. Subtitles from music apps covering the screen (users would close these)
  3. The pop-up implementation with "Please click: (x, y)" text is unrealistic
- The framework uses an additional LLM as onlooker for every action. What's the latency and cost impact? For practical deployment, this matters a lot.


Minor issues:
- Some figures are hard to read (Figure 2 is very crowded)
- The paper could better distinguish their work from GUI-Robust (Yang et al., 2025b) which seems similar

---

> ### Author Rebuttal · Authors · 2026-03-29
>
> Dear Reviewer G5fH,
>
> Thank you for your valuable comments. We will address your questions point by point as follows:
>
> >Q1. Contribution is engineering-focused......what new scientific insights we gain beyond agents fail when things are messy.
>
> We respectfully disagree with your assessment of the contributions of our work.
>
> 1. Regarding implementation complexity: We believe that **influential benchmarks are valued primarily for their practical utility and rigorous evaluation capabilities, rather than their implementation complexity**. Classic benchmarks such as ImageNet‑C[1]、 ImageNet-C-Bar[2] also emploies relatively simple perturbation. Although these methods are also simple, their superiority remains undiminished. Additionally, **AgentHijack achieves configurable analysis that was previously infeasible, providing a high-quality benchmark for systematically diagnosing vulnerabilities in GUI agents**, which is also a key contribution.
>
> 2. **Our work reveals three previously under-explored and scientifically meaningful insights:** (a)Vulnerable grounding capability. (b)Susceptible decisions. (c)Fail to perceive initial environment errors. These are actionable, diagnostic findings with clear interpretability, **rather than a trivial confirmation that “agents fail”.**
>
> 3. We further propose AgentHijack‑Agent, **which provides clear insights and directions for developing more robust and reliable agents in future work.**
>
> [1] Benchmarking Neural Network Robustness to Common Corruptions and Perturbations, ICLR 2019
>
> [2] On Interaction Between Augmentations and Corruptions in Natural Corruption Robustness, NeurIPS 2021
> >Q2. AgentHijack-Agent combines existing pieces. Contribution feels incremental.
>
> 1. **Our primary contribution is the benchmark,** which offers a systematic evaluation framework, the formal distinction between corruption and adversarial robustness, and fine-grained analyses of failure modes. AgentHijack-Agent serves as a fundamental baseline derived from various insights obtained through the benchmark, offering valuable directions for future researches.
>
> 2. AgentHijack-Agent is a targeted integration based on concrete diagnostic insights, **rather than a trivial or obvious combination.** Specifically, DA-GRPO addresses the poor grounding issue. The onlooker mitigates the problem where agents incorrectly attribute unexpected environmental changes to their own behaviors and the blind assumption that the initial state is always correct. The consistent and additive improvements demonstrate each component is indispensable, which provides insights for future works.
>
> >Q3. Some corruptions are unrealistic.
> 1. For marks: They are intended to simulate visual disruptors caused by common visual elements widely present in various operating systems and third-party applications[1], **such as marks like bubbles (circular) and ribbons (rectangular) in windows screensavers.**
>
> 2. For subtitle: While human users can manually remove subtitles when needed, such practice deviates from the original intention of autonomous agents. The reliance on a human-in-the-loop setting presupposes that humans can actively participate in task completion, which may undermine the necessity of deploying autonomous agents.
>
> 3. For pop-ups, “click (x,y)” is very common in advertisement pop-ups and software installation prompts, benchmarks on adversarial robustness also include this kind of corruption[2]. In addition, we would like to clarify that **all such text is configurable,** allowing users to freely customize corruptions that better suit their specific usage scenarios.
>
> [1] Visual Delegate Generalization Frame – Evaluating Impact of Visual Effects and Elements on Player and User Experiences in Video Games and Interactive Virtual Environments, CHI 2022
>
> [2] Attacking Vision-Language Computer Agents via Pop-ups, ACL 2025
> >Q4. Latency of onlooker?
>
> We measure latency of the onlooker as below.
> ||device|latency
> |-|-|-
> |summary|1*A100|1.73s
> |environment check|1*A100|1.42s
>
> Compared with the average total task latency (321.19s on an A100 GPU), the onlooker only **incurs an additional overhead of about 5%** (averaging 10 steps, with environment check executed only before the first step), while achieving an average performance improvement of 20%, which is acceptable.
>
> >Q5. Figure 2 is crowded.
>
> We will refine the figure to improve readability in the revised version.
>
> >Q6. distinguish our work from GUI-Robust.
>
> As illustrated in Table 1, GUI-Robust only employs **static question-answer pairs**, where agents are given corrupted screenshots and asked to predict the next correct action. This setup only supports single-turn interaction testing. Instead, our work provide **multi-turn interaction in realistic environment based on docker/VMVare**. Additionally, their screenshots are **fixed and non-configurable**. We also have more types of corruption instead of the similar corruptions in GUI-Robust such as advertisement pop-ups and cookie pop-ups.

---

> > ### Author Rebuttal · Reviewer_G5fH · 2026-03-31
> >
> > I thank the authors for their detailed rebuttal. The clarifications on latency overhead (Q4, ~5% for ~20% gain) and the distinction from GUI-Robust are convincing and address those concerns well. However, my core concern about significance (Q1) remains. The ImageNet-C analogy actually underscores the issue — ImageNet-C opened an entirely new research direction for DNN robustness, whereas this work applies an established evaluation paradigm (corruption benchmarking) to a new domain (GUI agents). The three insights mentioned (vulnerable grounding, susceptible decisions, failure to perceive initial errors) are reasonable findings but not particularly surprising — they largely confirm expected failure modes rather than revealing counter-intuitive behaviors. On Q2, I appreciate the honesty that the benchmark is the primary contribution, but this effectively confirms my initial concerns. Given all this, I will maintain my current score.

---

> > > ### Author Response · Authors · 2026-04-01
> > >
> > > Dear Reviewer G5fH,
> > >
> > > We are glad that our previous response has addressed some of your concerns. We would like to provide further clarification on the remaining issues as follows:
> > >
> > > Firstly, for the ImageNet-C analogy and novelty of the evaluation paradigm. We respectfully note that the transition from ImageNet-C to AgentHijack is not merely applying the same paradigm to a new domain. Instead, there is a fundamental methodological difference. ImageNet-C evaluates **single-step, feed-forward perception, where corruptions only affect the input.** In contrast, AgentHijack evaluates **multi-step, closed-loop interaction where corruptions affect observations, state transitions, and environmental initialization simultaneously.** This means that a single corruption can cascade through subsequent steps. For example, as shown in Figure 6, the same accidental touch corruption produces different failure patterns depending on when it occurs during execution. Such temporal and compounding effects do not exist in the ImageNet-C setting and cannot be studied without an interactive, executable environment. **This requires fundamentally different benchmark design considerations**, i.e., defining corruption scopes across observations, transitions, and environment states, ensuring corruptions interact properly with the execution loop, and supporting configurable parameters for systematic ablation. Therefore, we believe AgentHijack makes a meaningful methodological contribution beyond simply transferring the corruption benchmarking idea to a new domain.
> > >
> > > Secondly, for whether the findings are expected. We wish to draw a distinction between **general directional intuitions** and **specific, actionable mechanistic understandings.** One might intuitively expect that system performance degrades under external perturbations, much like how people could roughly anticipate that deep neural networks are sensitive to input noise prior to the introduction of ImageNet-C. However, **the specific failure mechanisms we uncover are concrete, reproducible, and not readily predictable from first principles.** For instance: agents tend to attribute state changes caused by external factors to their own actions, rather than recognizing the presence of interference. They incorporate this misattribution into their reasoning and proceed based on flawed assumptions. Agents also exhibit a strong bias toward assuming correct initialization, even in the face of clear error signals such as login pop-ups or network disconnections, leading to prolonged meaningless exploration. **These are far from generic observations that “agents fail.”** Instead, **they directly inform the design of more robust agent frameworks in future work.** This is not achievable by prior studies such as GUI-Robust, which rely on static QA-based evaluation and cannot capture concrete failure modes. We therefore believe that **AgentHijack provides valuable insights for future research on agent robustness.**
> > >
> > > Thirdly, for whether our contribution is merely incremental, we respectfully argue that the significance of a benchmark should be measured by the **new research directions it enables**, rather than **the technical complexity of its individual components.** Prior to the introduction of AgentHijack, researchers were unable to systematically answer critical questions such as: **How does different corruptions affect agent performance? Which failure modes require architectural changes versus training improvements?** Our configurable and executable framework enables all such systematic investigations for the first time. We allow researchers to conduct fine-grained analysis on different types of corruption, varying in intensity (Figure 4), content (Figure 5), and location (Figure 6). The analysis of concrete failure patterns presented in this paper, together with the proposed AgentHijack-Agent framework, **establishes a research infrastructure that can support a sustained line of future work on agent robustness**, which is much like ImageNet-C has done for the robustness of visual models. **Accordingly, we believe this enabling capability represents a major, rather than incremental, contribution.**

---

### Decision · Program_Chairs · 2026-04-30

**Decision:**

Accept (regular)

**Comment:**

This paper introduces AgentHijack, a benchmark dataset for evaluating robustness of computer use agents to environmental corruptions, such as pop-up dialogs and resolution changes. This benchmark introduces nine common corruptions to replicate real world scenarios. Authors evaluate different desktop tasks that utilize MLLM-based agents and show that such corruptions can significantly degrade performance of such agents. Authors also propose a solution through their introduced framework AgentHijack-Agent and show the effectiveness of it.

Reviewers appreciated the practical use-case of this work and the extensiveness of the proposed corruption taxonomy. They all appreciated the fact that the corruption taxonomy is realistic and grounded for real world scenarios. One reviewer raised concerns about the significance of this work and thought that the paper demonstrates a more incremental contribution. However, all other reviewers appreciated the work and agreed that it is timely, systematic, configurable, important, and provides value to the community. There was also a concern on effectiveness of the introduced framework in improving the performance of the agents to the introduced corruptions that remains partially unresolved by a reviewer which required additional ablation studies. However, the reviewer was positive about the contributions of the paper and thought that it has merit and value to be published.

The authors provided a thorough rebuttal with substantial additional experiments which addressed the majority of reviewer concerns except a partial resolution on the ablation studies for one reviewer and a major disagreement on contribution of the work by another reviewer.